# AvatarSync: Rethinking Talking-Head Animation through Phoneme-Guided Autoregressive Perspective

## Abstract

Talking-head animation focuses on generating realistic facial videos from audio input. Following Generative Adversarial Networks (GANs), diffusion models have become the mainstream, owing to their robust generative capability. However, inherent limitations of the diffusion process often lead to inter-frame flicker and slow inference, hindering their practical use in talking-head animation. To address this, we introduce AvatarSync, an autoregressive framework on phoneme representations that generates realistic and controllable talking-head animations from a single reference image, driven by text or audio input. To mitigate flicker and ensure continuity, AvatarSync leverages an autoregressive pipeline that enhances temporal modeling. In addition, to ensure controllability, we introduce phonemes that are the basic units of speech sounds, and construct a many-to-one mapping from text/audio to phonemes, enabling precise phoneme-to-visual alignment. To further accelerate inference, we adopt a two-stage generation strategy that decouples semantic modeling from visual dynamics, incorporating a Phoneme-Frame Causal Attention Mask and a timestamp-aware adaptive strategy to support parallel inference. Extensive experiments conducted on Chinese (CMLR) and English (HDTF) benchmarks show that AvatarSync substantially reduces inter-frame flicker and outperforms existing methods in visual fidelity, temporal consistency, and computational efficiency, providing a scalable solution.

## 1 Introduction

Talking-head animation Guo et al. (2024); Hu (2024); Tian et al. (2024b); Chen et al. (2025); Meng et al. (2024); Lu et al. (2021); Wei et al. (2024); Chu et al. (2025); Zhen et al. (2025); Wang et al. (2025) is a representative multimodal generation task that demands fine-grained alignment between audio and visual outputs. Leveraging advancements in artificial intelligence, this technique synthesizes realistic, speech-synchronized facial motion from static images and audio inputs. This technology finds widespread applications in areas such as video dubbing, virtual avatars, and digital entertainment Prajwal et al. (2020). Despite significant progress, efficiently generating high-quality, lifelike, and fine-grained talking-head animations in real time remains a formidable challenge.

In the field, two primary paradigms have emerged: Generative Adversarial Networks (GANs) Goodfellow et al. (2020) and diffusion models Ho et al. (2020). GAN-based methods Zhen et al. (2023); Cheng et al. (2022); Wang et al. (2023); Zhang et al. (2023b;a) offer advantages in inference speed and computational efficiency. However, they often suffer from visual artifacts and struggle to maintain identity consistency, limiting their applicability in high-fidelity scenarios. Recently, diffusion models Rombach et al. (2022); Wang et al. (2024a); Ji et al. (2024); Lin et al. (2025); Li et al. (2024); Jiang et al. (2024) have gained attention due to their superior visual fidelity in image generation tasks. Several works (such as EMO Tian et al. (2024b), Hallo Xu et al. (2024a); Cui et al. (2024), and EchoMimic Chen et al. (2025) Meng et al. (2024)) have extended diffusion models to talking-head animation. These approaches generally produce clearer and more stable visual results. Nonetheless, the reliance of diffusion models on multi-step denoising processes leads to slow inference and high computational cost, which severely hinders their deployment in real-time applications. To address these issues, recent efforts Ji et al. (2024); Li et al. (2024) have explored strategies, such as sampling path control, to improve inference efficiency. However, diffusion-based approaches still

Figure 1: Comparison of GAN-based, diffusion-based, and our autoregressive method. The left and middle panels summarize key limitations of GAN and diffusion models. The right panel illustrates the advantages of our autoregressive method.

suffer from fundamental limitations, including inter-frame flicker, unnatural facial dynamics, and poor real-time performance. As illustrated in Figure 1, although both GAN-based and diffusion-based methods have made significant progress, achieving a better balance among computational efficiency, generation consistency, and visual fidelity remains a major challenge in this field.

To this end, we introduce AvatarSync, an autoregressive framework on phoneme representations that generates realistic and controllable talking-head animations from a single reference image, driven by text or audio input. As illustrated in Figure 3, AvatarSync adopts a two-stage generation strategy, combining a Facial Keyframe Generation (FKG) module with the inter-frame interpolation module to synthesize natural facial dynamics. In the first stage, by leveraging this many-to-one relationship, the FKG module extracts character-level phoneme sequences from text or audio input. Subsequently, the phoneme sequences and reference image are respectively tokenized using a text tokenizer Ding et al. (2021) and a visual tokenizer trained with either VQ Van Den Oord et al. (2017); Esser et al. (2021) or LFQ Yu et al. (2023). These phoneme and visual tokens are then aligned and concatenated into a unified sequence, enabling an autoregressive transformer model to produce a sparse set of keyframes under a Phoneme-Frame Causal Attention Mask.

In the second stage, we propose a timestamp-aware adaptive strategy built upon a selective state space model, to enable efficient temporal modeling and precise audio-visual alignment. The interpolation module leverages explicit timestamp information embedded in keyframes to flexibly control motion intensity across variable frame intervals. In addition, to facilitate global context aggregation, adjacent keyframes are encoded as interleaved token sequences and processed through state space modeling. As a result, the system synthesizes natural and temporally coherent facial dynamics.

To support practical deployment, we structurally optimize the inference pipeline to significantly improve computational efficiency without compromising generation quality. AvatarSync outperforms conventional systems in most real-world scenarios, delivering a smooth and responsive user experience. Notably, AvatarSync establishes a new modeling paradigm and methodological framework for talking-head multimodal generation task. In summary, our main contributions are listed as follows:

- To substantially reduce inter-frame flicker, we propose AvatarSync, an autoregressive framework on phoneme representations that generates talking-head animations from a single reference image, driven by text or audio. By leveraging the many-to-one mapping from text/audio to phonemes, we construct phoneme-to-visual alignment. This design enables it to support editable, segment-level, and fine-grained control over video generation.

- We introduce a two-stage hierarchical generation strategy that decouples semantics from visual dynamics. The first stage, Facial Keyframe Generation (FKG), models phoneme-aligned semantics, while the second stage interpolates intermediate frames to enhance temporal coherence and visual smoothness. This design mitigates error accumulation, supports localized editing, and enables parallel inference for improved efficiency.

- In FKG, we design a Phoneme-Frame Causal Attention Mask to enhance phoneme-frame alignment and employ a composite loss integrating perceptual, identity, and facial similarity. For interpolation, we propose a timestamp-aware adaptive strategy based on selective state space modeling, enabling temporal inference and audio-visual synchronization.

- We conduct comprehensive evaluations of AvatarSync on two benchmark datasets, CMLR and HDTF, covering Chinese and English. As shown in Table 1 and Figure 10, AvatarSync consistently outperforms existing advanced audio-driven talking-head animation models in terms of computational efficiency, facial fidelity, and motion consistency.

## 2 RELATED WORK

### 2.1 TALKING HEAD GENERATION

Audio-driven talking-head generation has emerged as a key research topic in multimodal content generation, demonstrating significant practical value in applications such as video dubbing and virtual avatars. Prevailing approaches can be broadly categorized into two classes: GAN-based methods Zhou et al. (2019; 2021); Meshry et al. (2021); Das et al. (2020); Chen et al. (2019); Zhang et al. (2023a) and diffusion-based methods Wang et al. (2024a); Ji et al. (2024); Lin et al. (2025); Li et al. (2024); Jiang et al. (2024); Xu et al. (2024b). In the following, we provide a systematic review of recent advances and representative characteristics of each class.

**GAN-based methods.** GAN-based methods are widely recognized for their computational efficiency and rapid inference. However, early approaches struggle with maintaining identity consistency and accurate lip synchronization. To address this, methods, such as SadTalker Zhang et al. (2023a) and FaceVid2Vid Wang et al. (2021), adopt multi-stage inference pipelines that decouple audio-to-motion and motion-to-video modeling. While this improves generation quality, it significantly increases computational overhead and system complexity. Moreover, the decoupled modeling leads to unnatural generation results, where only the mouth moves while the rest of the face remains static, compromising realism and temporal continuity.

**Diffusion-based methods.** Diffusion-based approaches typically integrate ReferenceNet, temporal modeling layers, and audio-attention modules into a single unified framework. These methods enable vivid talking head generation from a single image, but come with high computational costs and often suffer from unstable mouth motion. To reduce the overhead, MuseTalk Zhang et al. (2024b) combines diffusion with GANs. OmniHuman-1 Lin et al. (2025) further proposes a hybrid training scheme based on a Diffusion Transformer architecture. While these methods partially alleviate slow inference and low visual fidelity, they do not overcome diffusion's inherent limitations, leaving generated videos with artifacts such as ghosting and inter-frame flicker.

### 2.2 VISUAL GENERATION BASED ON LARGE LANGUAGE MODELS

In recent years, large language models (LLMs) Achiam et al. (2023); Touvron et al. (2023); Liang et al. (2024) have extended to the domain of visual content generation. Compared to diffusion models that rely on multi-step denoising, LLM-based visual generation methods offer superior scalability and inference efficiency for multimodal tasks.

LLM-based visual generation approaches can be broadly categorized into two types: **masked language models (MLMs)** and **autoregressive language models (AR-LMs)**. MLMs enable efficient training and fast sampling by predicting randomly masked tokens in parallel. In image generation, MaskGIT Chang et al. (2022) progressively refines images by predicting missing tokens, achieving both high quality and computational efficiency. Subsequently, this approach is extended to the video domain. MAGVIT-v2 proposes an embedding method for iterative masked video token modeling.

AR-LMs predict tokens sequentially, modeling the conditional probability of each token given its preceding context. In image synthesis, LlamaGen Sun et al. (2024) employs an autoregressive Transformer to generate semantically aligned, detail-rich images, while VAR Tian et al. (2024a) adopts a coarse-to-fine generation strategy to iteratively refine multi-scale representations. In video generation, VideoPoet Kondratyuk et al. (2023) processes multimodal inputs through region-wise tokenization. CogVideo Hong et al. (2022), Show-o Xie et al. (2024) and EMU3 Wang et al. (2024b) further extend autoregressive modeling to text-to-video generation, proposing multimodal architectures. Recently, a few studies Chu et al. (2025); Zhen et al. (2025) have also explored transformer-based architectures specifically for talking-head generation, achieving promising results in both visual realism and controllability. These methods directly encode audio sequences and reference images using standard Transformer architectures to generate video frames. However, these methods typically entail high computational demands and inference complexity, limiting practical deployment.

Furthermore, as LLMs Khanuja et al. (2024); Fang et al. (2024); Shahmohammadi et al. (2023) are increasingly adopted in natural language processing, token-level parallelization strategies for accelerating autoregressive inference have rapidly gained traction in visual generation Leviathan et al. (2023); He et al. (2024); Fu et al. (2024). These approaches require no model retraining, offering

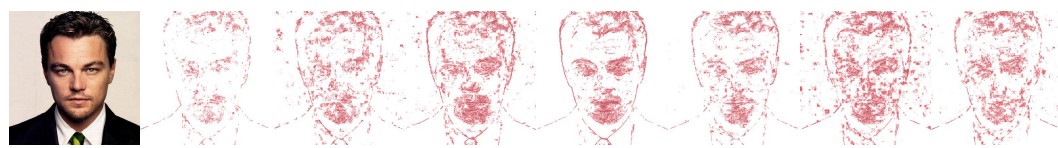

Figure 2: Inter-frame Flicker Visualization. Left: reference frame; subsequent panels show pixel-wise differences between consecutive frames, where scattered high-difference regions reveal temporal flicker.

strong generalizability and deployment flexibility. In summary, LLM-based methods represent a promising direction for achieving real-time, high-fidelity, and controllable talking-head generation.

## 3 METHOD

### 3.1 PRELIMINARY OF INTER-FRAME FLICKER

As illustrated in Figure 2, diffusion-based video generation methods often exhibit inter-frame flicker, manifesting as temporal inconsistencies or identity shifts between adjacent frames. In the following, we provide a theoretical analysis based on Denoising Diffusion Probabilistic Models formulation. Consider the DDPM reverse process for generating a single image frame $\hat{\mathbf{x}}_0^{(t)}$ from Gaussian noise:

$$\mathbf{x}_T^{(t)} \sim \mathcal{N}(\mathbf{0}, \mathbf{I}), \quad \hat{\mathbf{x}}_0^{(t)} = f_\theta(\mathbf{x}_T^{(t)}, \mathbf{c}^{(t)}) \tag{1}$$

where $t$ indexes the frame index, $\mathbf{c}^{(t)}$ is the conditioning input, and $f_\theta(\cdot)$ denotes the denoising trajectory defined by the model. Even under fixed $\mathbf{c}^{(t)} = \mathbf{c}$ across all frames, the sampled latent variables $\mathbf{x}_T^{(t)}$ are independent:

$$\text{Cov}(\mathbf{x}_T^{(t)}, \mathbf{x}_T^{(t+1)}) = \mathbf{0} \tag{2}$$

As a result, the output frames $\hat{\mathbf{x}}_0^{(t)}$ and $\hat{\mathbf{x}}_0^{(t+1)}$ are conditionally uncorrelated, resulting in inter-frame variability. Formally, the output distribution is:

$$p_\theta(\hat{\mathbf{x}}_0^{(t)}|\mathbf{c}) = \int p_\theta(\hat{\mathbf{x}}_0^{(t)}|\mathbf{x}_T^{(t)}, \mathbf{c}) \cdot \mathcal{N}(\mathbf{x}_T^{(t)}; \mathbf{0}, \mathbf{I}) \, d\mathbf{x}_T^{(t)} \tag{3}$$

Since $\mathbf{x}_T^{(t)}$ and $\mathbf{x}_T^{(t+1)}$ are independently and identically sampled from the standard Gaussian prior, adjacent frames are marginally independent even under identical conditioning. Consequently, the generated frame sequence $\{\hat{\mathbf{x}}_0^{(t)}\}_{t=1}^T$ is prone to exhibit a lack of temporal coherence.

While some diffusion models, such as DDIM Song et al. (2020), DiT Peebles & Xie (2023), and models employing 3D convolutions Ho et al. (2022), have begun to model temporal dependencies in the denoising process, the independent sampling of the initial noise $\mathbf{x}_T^{(t)}$ for each frame still leads to insufficient temporal coherence in the generated videos. To mitigate this, guided noise injection methods Li et al. (2024) have been proposed. However, the inherent stochasticity of the initial noise poses a significant challenge to fully resolving the issue of inter-frame flickering.

Autoregressive models generate video frames as a single and unified token sequence. Let $X = \{x_1^{(1)}, ..., x_K^{(T)}\}$ denote a flattened sequence of $T$ video frames, where each frame contains $K$ discrete tokens. Here, $x_i$ denotes the $i$-th token in the flattened sequence, and $x_j^{(t)}$ refers to the $j$-th token in frame $t$. The model estimates Ashish (2017):

$$P(X) = \prod_{i=1}^N P(x_i|x_{<i}, c) \tag{4}$$

For any token $x_j^{(t)}$ in frame $t$, its generation depends on all tokens from previous frames and prior tokens within the same frame:

$$P(x_j^{(t)}|x_1^{(1)}, ..., x_K^{(t-1)}, x_1^{(t)}, ..., x_{j-1}^{(t)}, c) \tag{5}$$

Here, when $t = 1$ or $j = 1$, the corresponding conditioning sets are empty. Therefore, compared to diffusion models, autoregressive models generate frames sequentially with strong contextual conditioning, exhibiting a strong inductive bias toward temporal coherence.

## 3.2 MODEL DESIGN

The overall framework of AvatarSync is depicted in Figure 3. It mainly consists of three parts: (1) an image tokenizer for quantizing the reference image into visual tokens, and an audio ASR tool for processing the input speech into a phoneme sequence; (2) a two-stage generation model based on an autoregressive framework, designed to effectively model phoneme-conditioned multimodal inputs and synthesize high-quality visual outputs; (3) a decoder for performing downstream tasks. In the following, we focus on detailing the first and second components of the system.

**Tokenization.** AvatarSync is flexible for handling multimodal input, supporting both text-image and audio-image modalities for video generation. (1) *For text input*, the input text is first converted into a phoneme sequence, leveraging the stable many-to-one mapping to facilitate accurate mouth-shape generation. Subsequently, a standard tokenizer transforms the phoneme sequence into discrete tokens. (2) *For audio input*, we employ automatic speech recognition (ASR) tools to extract phoneme-level alignments with timestamps, which are then tokenized into discrete phoneme tokens. (3) *For an image input*, we adopt a pre-trained vision foundation model, such as Open-MAGVIT2 Luo et al. (2024), to extract image features. To capture fine-grained facial details, we employ MMPose for facial landmark detection and adjust the input image's aspect ratio.

**Auto-regressive Model.** Following prior work Yan et al. (2021); Kondratyuk et al. (2023), text, audio and image prompts are projected into the feature space of a large language model (LLM). As illustrated in Figure 3, our autoregressive model follows a pipelined generation process consisting of two stages: Facial Keyframe Generation (FKG) and the inter-frame interpolation module. Notably, unlike the two-stage designs in prior work Harvey et al. (2022); Wei et al. (2024), our framework aligns keyframes explicitly with phoneme units rather than uniformly sampling in time, and the second stage interpolates frames within these phoneme-aligned intervals using their timestamps.

(1) The model generates $T_s$ keyframes in accordance with the sequential order of the input phoneme. The Facial Keyframe Generation (FKG) module receives phoneme representations encoded by a tokenizer and structures the input sequence as: $\{\{\text{Phoneme}\}$ [B] $\{\text{Frame}_1\},$ ..., $\{\text{Frame}_{T_s}\}\}$. In addition, we introduce a Phoneme-Frame Causal Attention Mask, which restricts each keyframe to its paired phonemes and masking cross-frame attention to avoid leakage. Specifically, when generating each keyframe, the model attends only to its corresponding phoneme information, enabling precise phoneme-to-frame mapping and temporally aligned phoneme modeling. In practice, the model conditions on both phoneme information and the reference image, and employs a parallel strategy to simultaneously predict $T_s$ keyframes.

(2) The interpolation module operates on phonemes, timestamps, and known keyframes. Drawing on VFIMamba Zhang et al. (2024a), we introduce a timestamp-aware adaptive strategy built upon a selective state space model, enabling efficient temporal modeling and precise audio-visual alignment. Specifically, guided by phoneme-timestamp pairs, intermediate frames are inserted between keyframes. Additionally, at each interpolation step, adjacent keyframes are encoded into interleaved token sequences and processed via state space modeling, enabling efficient global context aggregation with linear complexity. This design progressively refines frame durations based on phoneme rhythm, ensuring temporal coherence, synchronization with audio, and stable output frame rates. Furthermore, interpolations between different keyframe pairs can be performed in parallel, significantly improving inference efficiency.

## 3.3 FACIAL TRAINING STRATEGY

In training AvatarSync, we decouple semantic accuracy from visual refinement. The FKG module is optimized for semantic precision, while the interpolation module focuses on temporal coherence and visual smoothness. For the FKG training, we employ a composite loss function that integrates reconstruction, perceptual similarity, identity preservation, and facial appearance fidelity. To mitigate the instability caused by simultaneous optimization of multiple objectives, we adopt a phased training strategy. The training objective of the first stage is to learn abstract facial inpainting using a single loss function:

$$\mathcal{L}_{recon} = -\sum_i \log P\left(v_i^{real} \mid \mathbf{x}\right) \tag{6}$$

where $v_i^{real}$ is the ground-truth token at position $i$, and $P(\cdot)$ represents the predicted probability distribution over the token vocabulary. In the second stage of training, we operate in the decoded

Figure 3: The overall framework of AvatarSync. The pipeline first normalizes text/audio into a phoneme token sequence via a many-to-one mapping, and tokenizes the reference image into visual tokens. Next, a two-stage autoregressive generator performs Facial Keyframe Generation under a Phoneme-Frame Causal Attention Mask, and inserts intermediate frames using a timestamp-aware module that interleaves keyframes for linear-time global context. Finally, the decoder reconstructs RGB frames to animate character.

pixel space and incorporate three loss terms: LPIPS Perceptual loss Li et al. (2024); Zhang et al. (2018), Identity Consistency loss, and Facial Similarity loss to enhance visual quality.

$$\mathcal{L}_{lpips} = \sum_l w_l \cdot \frac{1}{H_l W_l} \sum_{h,w} \|F_l(I_{\text{gen}})_{h,w} - F_l(I_{\text{real}})_{h,w}\|_2^2 \tag{7}$$

where $F_l(\cdot)$ denotes the feature map from the layer $l$, and $H_l$, $W_l$ are its height and width.

$$\mathcal{L}_{Id} = \frac{1}{N} \sum_{i=1}^{N} \left(1 - \cos(f_{gen}^i, f_{real}^i)\right) \cdot w_{id} \tag{8}$$

where $f_{gen}^i$ and $f_{real}^i$ are identity embeddings of the $i$-th generated and real image, respectively.

$$\mathcal{L}_{FS} = \frac{1}{N} \sum_{i=1}^{N} 0.5 \cdot d_{cos}(f_{gen}^i, f_{real}^i) \cdot w_{fs} \tag{9}$$

where $d_{cos}(\cdot, \cdot)$ measures the cosine distance in the FaceNet512 embedding space. The overall optimization objective for this stage is:

$$\mathcal{L}_{total} = \lambda_1 \cdot \mathcal{L}_{recon} + \lambda_2 \cdot \mathcal{L}_{lpips} + \lambda_3 \cdot \mathcal{L}_{Id} + \lambda_4 \cdot \mathcal{L}_{FS} \tag{10}$$

## 3.4 DATA PREPARATION

To support keyframe generation, we construct two phoneme-to-frame aligned training datasets: the Chinese Mandarin Lip Reading (CMLR) dataset Zhao et al. (2019; 2020) and the English-speaking HDTF dataset Zhang et al. (2021), enabling cross-lingual modeling. Initially, we apply ASR tools to extract phonemes and their corresponding timestamps from the audio tracks, and use them to retrieve the aligned video frames. Facial regions are then detected and cropped to obtain phoneme-aligned face images. In addition, due to the low resolution of CMLR, we incorporate the GFPGAN face enhancement algorithm to perform four-times super-resolution reconstruction. To reduce encoder training complexity, we further map abstract phonemes to concrete, character-level units. For coarticulation, we explicitly introduce special mappings for specific coarticulation patterns (e.g., two characters are jointly pronounced). This preprocessing results in two phoneme-image paired datasets. Notably, we compare two strategies for facial region extraction: Face-Centric Cropping and Pose-Driven Landmark Cropping. Based on empirical results (see Table 4 in Appendix A.3), we adopt the pose-driven method.

Table 1: Quantitative comparison on CMLR and HDTF datasets. SadTalker is GAN-based, the other baselines are diffusion models, and AvatarSync (ours) is autoregressive.

| Method | CMLR | | | | | | | | HDTF | | | | | | | |
|---|---|---|---|---|---|---|---|---|---|---|---|---|---|---|---|---|
| | FID↓ | FVD↓ | LPIPS↓ | PSNR↑ | SSIM↑ | CSIM↑ | Sync-C↓ | Sync-D↓ | FID↓ | FVD↓ | LPIPS↓ | PSNR↑ | SSIM↑ | CSIM↑ | Sync-C↓ | Sync-D↓ |
| SadTalker | 30.96 | 618.74 | 0.42 | 14.50 | 0.48 | 0.79 | 0.91 | 14.29 | 37.20 | 316.78 | 0.26 | 18.85 | 0.74 | 0.82 | 5.72 | 11.83 |
| V-Express | 32.54 | 821.95 | 0.46 | 12.99 | 0.39 | 0.70 | 0.83 | 13.71 | 40.13 | 779.19 | 0.27 | 18.29 | 0.72 | 0.88 | 5.29 | 12.79 |
| AniPortrait | 32.46 | 711.30 | 0.47 | 14.71 | **0.49** | 0.69 | 0.89 | 14.99 | 40.19 | 614.01 | 0.28 | 18.52 | 0.72 | 0.88 | 5.61 | 11.30 |
| Hallo | 30.84 | 630.53 | 0.43 | 14.62 | **0.49** | 0.71 | 1.20 | 14.99 | 44.48 | 520.72 | 0.25 | 18.94 | **0.75** | 0.88 | 5.05 | 11.62 |
| Hallo2 | 35.10 | 570.12 | 0.44 | 14.59 | **0.49** | 0.75 | **0.80** | 14.67 | 42.77 | 340.83 | 0.25 | **19.04** | **0.75** | 0.89 | 6.37 | 11.91 |
| EchoMimic | 33.09 | 1225.10 | 0.42 | 14.87 | 0.48 | 0.69 | 0.89 | 13.97 | 38.11 | 301.33 | 0.29 | 18.11 | 0.73 | 0.85 | 2.63 | 12.88 |
| Sonic | 31.58 | 953.30 | 0.43 | 14.39 | 0.47 | 0.68 | 2.25 | 12.84 | 32.89 | 379.13 | 0.26 | 18.55 | 0.74 | 0.90 | 7.65 | 12.70 |
| StableAvatar | 39.38 | 741.36 | 0.46 | 14.41 | **0.49** | 0.79 | 0.92 | 14.83 | 35.39 | 700.82 | 0.28 | 18.41 | 0.72 | 0.85 | 5.34 | 12.56 |
| **AvatarSync** | **25.19** | **503.29** | **0.40** | **15.07** | **0.49** | **0.85** | 0.94 | **10.90** | **29.72** | **260.65** | **0.24** | 18.69 | 0.73 | **0.95** | 2.57 | **10.39** |

## 4 EXPERIMENTS

### 4.1 EXPERIMENTAL SETUP

**Training Details.** We train AvatarSync on a mixed dataset that combines the super-resolved Chinese CMLR dataset and the original English HDTF dataset, with a standard 95:5 train-test split applied to each benchmark before mixing. The training is conducted for a total of 10,000 steps on this mixed dataset, using a total of 8 NVIDIA V100 and 2 NVIDIA L20 GPUs. At the core of the training stage, we introduce a custom Phoneme-Frame Causal Attention Mask and utilize a meticulously designed composite loss function to fine-tune the pre-trained model weights. For optimization, we employ the Adam optimizer with a learning rate of $2 \times 10^{-4}$, complemented by a cosine annealing schedule. To ensure memory efficiency, we enable 16-bit mixed-precision training, accelerated by the DeepSpeed ZeRO-2 framework. The complete training procedure is detailed in Appendix A.2.

**Evaluation Metrics.** We evaluate the models with eight complementary metrics. For perceptual quality and identity preservation, FID, FVD, LPIPS, and CSIM are computed in deep feature space, where lower FID/FVD/LPIPS and higher CSIM are better. For frame-level fidelity, PSNR and SSIM assess reconstruction accuracy with respect to the ground-truth frames in pixel space (higher is better). Finally, Sync-C and Sync-D quantify audio–visual lip synchronization (lower is better).

**Compared Baselines.** We compare AvatarSync with SOTA audio-driven talking-head methods, including both GAN-based and diffusion-based approaches. For GAN-based models, we consider SadTalker Zhang et al. (2023a). Diffusion-based baselines include V-Express Wang et al. (2024a), AniPortrait Wei et al. (2024), Hallo Xu et al. (2024a), Hallo2 Cui et al. (2024), EchoMimic Chen et al. (2025), Sonic Ji et al. (2024), and StableAvatar Tu et al. (2025). These models leverage various strategies such as multimodal attention, hierarchical diffusion, and landmark/audio conditioning.

### 4.2 QUANTITATIVE EVALUATION

**Comparison on CMLR and HDTF.** As shown in Table 1, AvatarSync consistently achieves state-of-the-art performance on both the Chinese CMLR and English HDTF datasets. Specifically, on CMLR, it achieves the best scores in generation realism and temporal coherence (FID, FVD). The state-of-the-art FVD score provides direct quantitative evidence of suppressed inter-frame flicker. Furthermore, it also matches or surpasses all baselines in reconstruction metrics (LPIPS, PSNR, SSIM) and identity similarity (CSIM), achieving the lowest Sync-D and competitive Sync-C. On HDTF, AvatarSync achieves state-of-the-art performance in six key metrics, which indicates accurate identity preservation and precise lip

Table 2: Inference efficiency on 3.9s clips at $512 \times 512$ resolution.

| Method | Latency(s)↓ | RTF↓ |
|---|---|---|
| SadTalker | 78.90 | 20.13 |
| V-Express | 167.90 | 42.83 |
| AniPortrait | 575.50 | 146.81 |
| Hallo | 444.30 | 113.34 |
| Hallo2 | 427.50 | 109.06 |
| EchoMimic | 512.70 | 130.79 |
| Sonic | 190.40 | 48.57 |
| StableAvatar | 187.00 | 47.70 |
| **AvatarSync** | **58.00** | **14.80** |

synchronization. These results demonstrate that AvatarSync can generate high-quality talking-head videos and exhibits strong generalization ability across languages.

**Inference Efficiency.** In AvatarSync, the inter-frame module processes each keyframe pair in parallel, reducing the complexity from $\mathcal{O}\big(\sum_i L_i\big)$ to $\mathcal{O}\big(\max_i L_i\big)$, where $L_i$ is the length of interval $i$. With $T_s$ roughly uniformly spaced keyframes, this design yields up to a $(T_s - 1)\times$ theoretical speedup over a naive generator and supports multi-GPU deployment. To further evaluate the in-

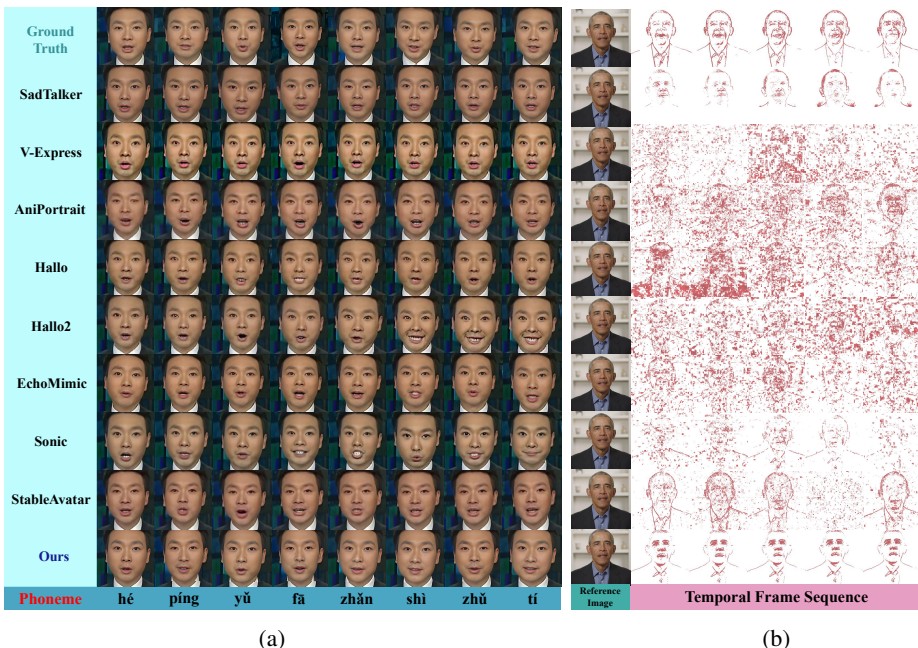

(a)                    (b)

Figure 4: Qualitative comparison on the CMLR and HDTF dataset. (a) Top: ground-truth frames. Middle: results from baseline models. Bottom: Each phoneme (represented as pinyin for Chinese) is aligned with its corresponding frame. (b) Inter-frame flicker visualization, where pixel-wise differences between consecutive frames highlight temporal inconsistencies across methods.

ference efficiency of AvatarSync, we compare it with existing GAN and diffusion-based methods. Under 3.9s clips at $512 \times 512$ resolution, Table 2 shows that AvatarSync achieves the lowest latency (58.0s) and real-time factor (RTF 14.8). As shown in Figure 10 in Appendix A.8, our model's latency exhibits a near-linear relationship with the input phoneme count, while competitors show exponential scaling. The parallelizable design makes our model practical for long talking-head videos.

## 4.3 QUALITATIVE EVALUATION

Qualitative comparisons in Figure 4a reveal two primary failure modes in existing methods. First, methods such as SadTalker, AniPortrait, and StableAvatar produce blurry reconstructions with imprecise lip articulation, while EchoMimic generates nearly static mouth shapes, all indicating weak audio-visual correlation. Second, others like V-Express, Hallo, Hallo2, and Sonic suffer from structural degradation, introducing warping artifacts in the lower face that render outputs unusable. In contrast, AvatarSync generates precise, dynamic mouth shapes that accurately track phonemes while preserving high-fidelity facial anatomy. This dual capability eliminates the articulatory imprecision, blurring, and distortion endemic to prior work, setting a new standard for realistic talking avatars.

Beyond per-frame quality, we evaluate temporal stability via inter-frame difference heatmaps in Figure 4b. The results indicate that diffusion-based methods exhibit severe and widespread flicker across the entire frame. In contrast, AvatarSync's pixel changes are minimal and strictly localized to the articulating mouth and jaw. This stability is inherent to autoregressive architecture, which conditions each frame on prior ones to enforce temporal coherence. The sequential dependency eliminates the stochastic variations that cause flicker, ensuring SOTA temporal coherence.

## 4.4 HUMAN EVALUATION

We conducted a human evaluation to assess the quality of generated animations. Thirty participants rated nine state-of-the-art methods across four dimensions: Flicker, body movement realism, temporal coherence, and lip synchronization, using a 5-point Likert scale. To ensure fair comparison, videos were randomly presented, providing insights into subjective perceptions of animation quality and natural expression alignment. As shown in Figure 9 in Appendix A.7, our method achieved the highest overall score, with strong scores across all dimensions (4.8 ± 0.41, 3.8 ± 0.48, 4.4 ± 0.50, 3.6 ± 0.62) and low variance in scores, which indicates robust and consistent performance.

### 4.5 ABLATION STUDIES

**Attention Mechanisms.** To validate the necessity of the Phoneme-Frame Causal Attention Mask, we conducted an ablation study on attention mechanisms using the CMLR dataset. Based on the scope of accessible phoneme, four distinct attention configurations

Table 3: Ablation of Attention Mechanisms on CMLR.

| ID | Setting | FID↓ | FVD↓ | LPIPS↓ | PSNR↑ | SSIM↑ | Sync-C↓ |
|---|---|---|---|---|---|---|---|
| 1 | Non-Causal | **15.63** | 287.95 | 0.07±0.01 | **24.41** | 0.86 | **1.21** |
| 2 | Causal Acc. | 19.47 | 210.25 | 0.07±0.01 | 23.35 | 0.86 | 1.24 |
| 3 | Lim. Hist. | 19.56 | **186.68** | 0.07±0.01 | 23.83 | 0.86 | 1.32 |
| 4 | One-to-One | 17.11 | 189.24 | 0.07±0.01 | 24.14 | 0.86 | 1.32 |

were compared: (1) Non-Causal Global Attention, (2) Causal Accumulative Attention, (3) Limited History Attention (sliding window size=2), (4) One-to-One Attention (Ours). As shown in Table 3, One-to-One Attention (Ours) achieves the optimal trade-off between frame-level fidelity and temporal coherence demonstrating the most robust overall performance. Detailed definitions of each mechanism are provided in Appendix A.4.

**Frame Allocation Strategies.** To verify the contribution of the timestamp-aware adaptive strategy, we compare three frame allocation strategies on CMLR: Random, Fixed, and Dynamic (ours). As summarized in Table 5 in Appendix A.5, the timestamp-aware adaptive strategy achieves the best performance on seven metrics, which confirms that it improves temporal coherence and lip synchronization without degrading reconstruction quality.

**Loss Components.** Our ablation study on four key loss terms: token-level cross-entropy (CE), pixel-level LPIPS, identity consistency, and facial similarity. Results (see Table 6 in Appendix A.6) show that while each component improves over the CE-only baseline, their combination yields consistently stronger performance. Notably, excluding identity or facial similarity losses leads to a marked drop in generation quality, highlighting their importance in preserving identity.

**Phoneme-based Representation Learning (PRL).** To assess whether phonemes are necessary as intermediate conditioning units, we compare two audio-driven configurations on the Chinese CMLR dataset: a baseline that conditions keyframe generation on audio features aligned with word-level units in the transcript, and a PRL configuration that uses the same audio aligned at the phoneme level. As shown in Fig. 5, phoneme-level conditioning respectively reduces the face reconstruction, total, and non-reconstruction losses by 41.8%, 9.6%, and 21.5%, indicating that phoneme-level conditioning provides finer-grained and more stable speech–lip alignment.

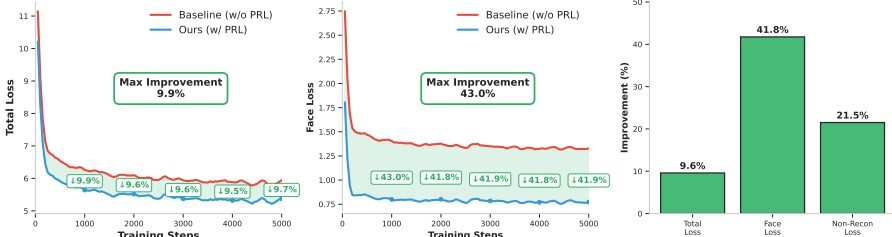

Figure 5: Loss comparison with and without PRL.

## 5 CONCLUSION

We introduce AvatarSync, an autoregressive framework on phoneme representations for talking-head animation generation. The method addresses two major limitations of diffusion-based approaches: (1) inter-frame flickers in generated videos; and (2) low training and inference efficiency. By leveraging the stable many-to-one mapping from text/audio to phonemes, AvatarSync enables accurate lip synchronization with lightweight design and editable controllability. To further improve temporal coherence and inference efficiency, we design a two-stage hierarchical generation strategy that decouples phoneme semantics from visual dynamics, incorporating a Phoneme-Frame Causal Attention Mask and a timestamp-aware interpolation module. Experimental results on the CMLR and HDTF datasets demonstrate that AvatarSync substantially reduces inter-frame flicker and consistently outperforms existing methods in visual fidelity, temporal consistency, and inference speed. Future work will leverage large-scale codebooks and MoE frameworks to achieve robust multilingual generalization, enabling a new generation of lifelike and interactive digital human applications.

ETHICS STATEMENT

In developing AvatarSync, a phoneme-guided autoregressive talking-head generation framework, we are committed to adhering to ethical principles and promoting responsible AI usage. We recognize potential risks, including deepfake abuse, impersonation, and unauthorized manipulation of personal media, and emphasize the necessity of applying this technology in contexts that respect privacy, consent, and individual rights.

REPRODUCIBILITY STATEMENT

To encourage transparency and responsible research, our code and pretrained models will be publicly released for academic and educational purposes, while we strongly discourage harmful applications such as misinformation, defamation, or harassment. Furthermore, we advocate for ongoing research on detection mechanisms and safeguard strategies to mitigate misuse, ensuring that AvatarSync contributes positively to society and aligns with ethical and legal standards.

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

# A APPENDIX

## A.1 DETAILS OF CMLR SUPER-RESOLUTION

The scarcity of high-quality, large-scale Chinese talking-head datasets poses a significant challenge to research in this domain. The CMLR dataset stands as one of the few publicly available Chinese datasets for this task, offering a crucial resource for research. However, its inherent low resolution results in blurry facial features and a lack of crucial detail in the lip region. This directly compromises the training efficacy and evaluation reliability of models that require high-fidelity visual input.

To address this limitation and establish a more robust benchmark, we employed the GFPGAN face enhancement algorithm to perform a comprehensive four-times super-resolution reconstruction across the entire CMLR dataset. A visual comparison of the frames before and after this enhancement is presented in Figure 6 and 7.

Furthermore, to foster future research and benefit the community, we will open-source this enhanced, high-resolution version of the CMLR dataset.

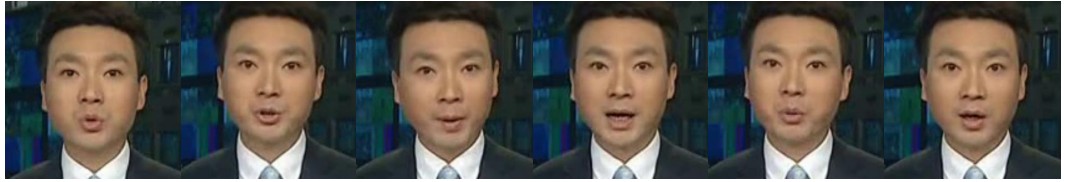

Figure 6: Original Video Frames from the Dataset.

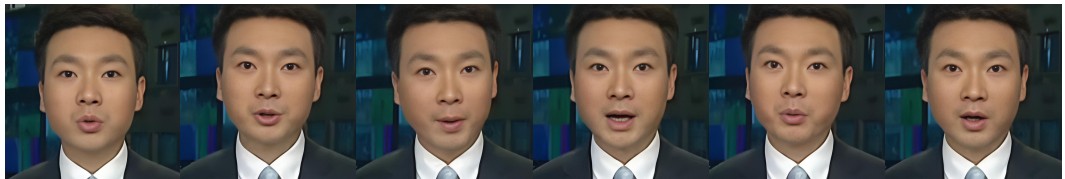

Figure 7: Enhanced Video Frames after Super-Resolution.

## A.2 TRAINING DETAILS

We trained the model on a mixed dataset that combines the super-resolved CMLR dataset (Chinese) and the original HDTF dataset (English). The training was conducted for a total of 10,000 steps on this mixed dataset.

Figures 11a, 11b, 11c, 11d, 11e, and 11f illustrate the progression of various loss functions during training, demonstrating the convergence behavior and the contribution of individual loss components to the total loss.

## A.3 FACE CROPPING STRATEGIES ABLATION DETAILS

**Cropping Strategy.** We compare two preprocessing methods: Face-Centric Cropping and Pose-Driven Landmark Cropping. The former leads to unstable generation due to scale and background variations. In contrast, the landmark-based approach ensures tighter alignment and better lip dynamics. In addition, the choice of face cropping strategy significantly impacts the final generation quality. Therefore, we conducted this ablation study to validate our choice of the Pose-Driven Landmark Cropping strategy over the baseline Face-Centric Cropping. Both qualitative and quantitative results confirm the superiority of our approach.

Qualitatively, as shown in Figure 8, our method yields tighter facial alignment and more consistent lip dynamics, resulting in enhanced visual coherence and identity preservation. Quantitatively, Table 4 shows our strategy yields a lower (better) Identity Similarity Score (ISS) between the generated faces and the ground-truth video on a majority of the face recognition models (3 out of 4).

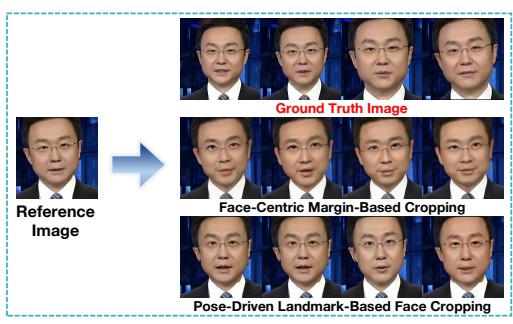 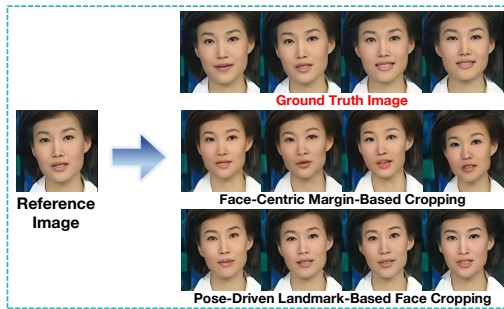

(a) Face-Centric Cropping  (b) Pose-Driven Landmark Cropping

Figure 8: **Visual comparison of face preprocessing methods.**

| Model / Subset | Face-Centric Cropping | | | | Landmark-Based Cropping | | | |
|---|---|---|---|---|---|---|---|---|
| | ArcFace | FaceNet | FaceNet512 | VGG-Face | ArcFace | FaceNet | FaceNet512 | VGG-Face |
| **s1** | 0.2958 | 0.1608 | 0.1931 | 0.2899 | 0.3250 | 0.2175 | 0.2360 | 0.3399 |
| **s2** | 0.2189 | 0.1672 | 0.1278 | 0.2885 | 0.2077 | 0.1886 | 0.1011 | 0.2236 |
| **s3** | 0.2576 | 0.1715 | 0.1079 | 0.2899 | 0.2873 | 0.1784 | 0.0752 | 0.2012 |
| **s4** | 0.3698 | 0.3415 | 0.2198 | 0.3643 | 0.3628 | 0.2822 | 0.1922 | 0.3465 |
| **s5** | 0.3137 | 0.2790 | 0.1677 | 0.3588 | 0.3015 | 0.1978 | 0.1319 | 0.2626 |
| **Total** | **0.2912** | 0.2240 | 0.1632 | 0.3183 | 0.2968 | **0.2129** | **0.1472** | **0.2748** |

Table 4: Identity similarity (ISS) comparison under different cropping strategies. Lower ISS values indicate greater identity similarity. **Bold numbers** in the **Total** row indicate better-performing cropping strategy per model.

Given its superior performance in both visual quality and quantitative identity preservation, we adopted the Pose-Driven Landmark Cropping strategy for all experiments.

### A.4 ATTENTION MECHANISMS ABLATION DETAILS

To validate the necessity and design rationale of our proposed Phoneme-Frame Causal Attention Mask, we conducted a key ablation study on the super-resolved CMLR dataset. We designed and compared four distinct attention configurations, which primarily differ in the scope of phonetic information accessible to the model during the generation of each frame. The details of these four attention mechanisms are as follows:

**(1) Non-Causal Full Attention.** When generating any frame, the model can access the entire input phoneme sequence from beginning to end. This configuration sees "future" information, making it unsuitable for streaming generation tasks. Its results are typically considered a theoretical performance upper bound.

**(2) Causal Accumulative Attention.** When generating the $i$-th frame, the model can access all historical phonemes from the 1st to the current $i$-th. This represents a standard autoregressive (causal) attention mechanism.

**(3) Limited History Attention.** When generating the $i$-th frame (for $i > 1$), the model utilizes a sliding window of size 2, accessing only the current $i$-th and the previous $(i-1)$-th phonemes. This strategy aims to provide limited local context while maintaining high computational efficiency.

**(4) One-to-One Attention.** When generating the $i$-th frame, the model strictly accesses only the corresponding $i$-th phoneme. This is the strictest form of causality, ensuring that the generation of each frame depends solely on the currently aligned input, without reliance on any historical or future information.

Table 5: Ablation of frame allocation strategies on CMLR.

| Method | FID↓ | FVD↓ | LPIPS↓ | PSNR↑ | SSIM↑ | CSIM↑ | Sync-C↓ | Sync-D↓ |
|---|---|---|---|---|---|---|---|---|
| Random | 24.82 | 553.39 | **0.40** | **15.07** | 0.49 | **0.85** | 1.09 | 14.11 |
| Fixed | **24.56** | 522.32 | 0.41 | 15.04 | 0.49 | 0.84 | **0.94** | 14.08 |
| Dynamic (ours) | 25.19 | **503.29** | **0.40** | **15.07** | 0.49 | **0.85** | **0.94** | **10.90** |

## A.5 Ablation on Frame Allocation Strategies

To further analyze the inter-frame module, we compare three frame allocation strategies while keeping the keyframe generator (FKG) and all other components fixed:

**(1) Random**. For each keyframe pair, the number of in-between frames is randomly sampled under the constraint that the total number of frames matches the target video length.

**(2) Fixed**. A fixed number of in-between frames is inserted for every keyframe pair, ignoring the actual temporal interval between them.

**(3) Dynamic (ours)**. The proposed timestamp-aware adaptive strategy allocates the number of in-between frames proportionally to the temporal interval between keyframes, while enforcing a globally consistent frame rate.

As shown in Table 5, the Dynamic strategy achieves clearly better temporal performance than Random strategy and Fixed strategy, with a lower FVD (503.29) and a markedly reduced Sync-D (10.90), while keeping FID, LPIPS, PSNR, SSIM, and CSIM at a comparable level. This confirms that the timestamp-aware adaptive allocation improves temporal coherence and phoneme-level lip synchronization without sacrificing reconstruction quality.

## A.6 Loss Function Ablation Details

To validate the effectiveness of each component in our proposed composite loss function, we conduct a detailed ablation study, with the full results presented in Table 6. In this study, we establish a baseline model trained exclusively with a token-level cross-entropy (CE) loss. We then incrementally incorporate our other proposed loss terms: the pixel-level LPIPS perceptual loss, identity consistency loss, and facial similarity loss.

The experimental results clearly demonstrate that while each loss component individually yields performance gains over the baseline, the optimal overall generation quality is achieved only through their combination. Particularly noteworthy is the finding that removing either the identity consistency or the facial similarity loss from the full model leads to a marked degradation in generation quality. This underscores their critical roles in preserving subject identity and enhancing visual realism.

## A.7 Human Evaluation Results

To assess subjective perceptual quality, we conducted a user study comparing AvatarSync with nine state-of-the-art audio-driven talking-head methods. Thirty participants were asked to rate the generated videos along four dimensions: Flicker, Temporal Coherence, Body Movement Realism, and Lip Synchronization, using a 5-point Likert scale (higher is better). For each audio clip, the videos from different methods were shuffled and anonymized to avoid ordering and naming bias.

Figure 9 reports the mean and standard deviation of the scores. AvatarSync achieves the highest average rating on all four dimensions ($4.8 \pm 0.41$ for Flicker, $3.8 \pm 0.48$ for Body Movement Realism, $4.4 \pm 0.50$ for Temporal Coherence, and $3.6 \pm 0.62$ for Lip Synchronization), consistently outperforming competing approaches with relatively low variance across subjects. These results corroborate the quantitative metrics, indicating that AvatarSync produces visually stable, natural, and well-synchronized talking-head videos from a human perspective.

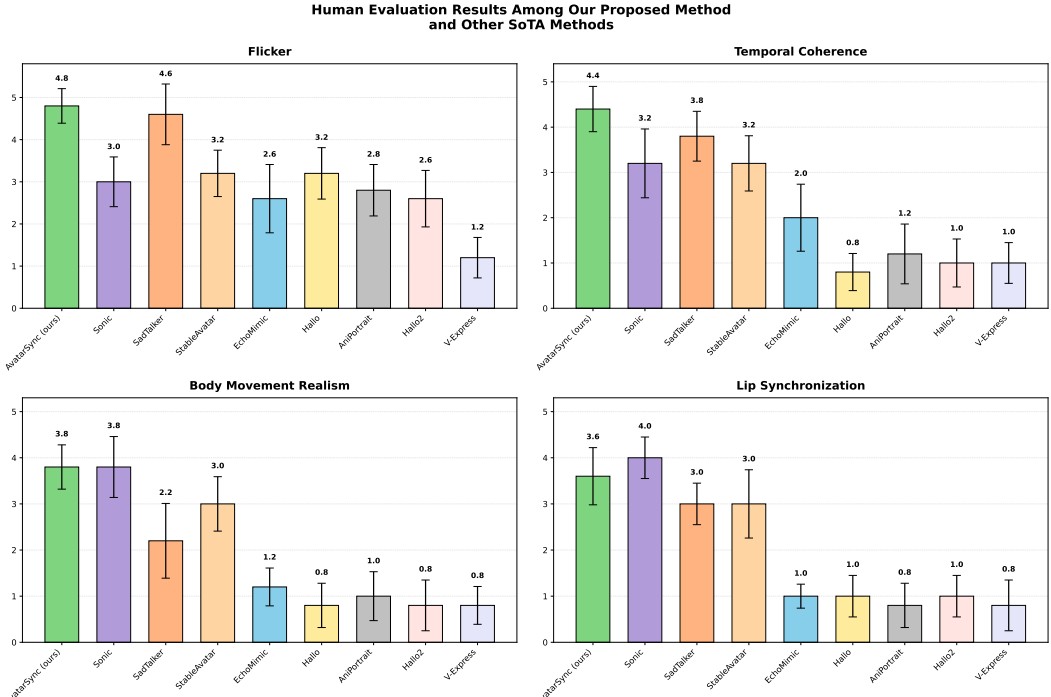

Figure 9: Human evaluation results on four aspects: Flicker, Temporal Coherence, Body Movement Realism, and Lip Synchronization. Bars show the mean opinion scores and error bars denote standard deviation over 30 participants. AvatarSync (ours) obtains the highest overall score, with strong performance across all dimensions.

## A.8 ANALYSIS OF INFERENCE EFFICIENCY

To complement the latency comparison in Table 2, we further visualize how inference time scales with the input length for different methods. Specifically, we vary the number of phonemes from 2 to 20 and measure the average generation time for $512 \times 512$ videos on the same hardware. Figure 10 shows that AvatarSync exhibits an almost linear growth with respect to the phoneme count, whereas several diffusion-based baselines grow super-linearly and quickly become impractical at longer durations. For example, at 20 phonemes, AvatarSync is about $2.4\times$ faster than Hallo and remains the most efficient among all competing methods. This is consistent with the theoretical parallelism analysis in Sec. 4.2.

## THE USE OF LARGE LANGUAGE MODELS(LLMS)

We utilized a large language model as a general-purpose writing assistant during the preparation of this paper. Its role was strictly limited to improving grammar, spelling, and overall language clarity. The authors are fully responsible for the research ideation, data, analysis, and final content of this manuscript.

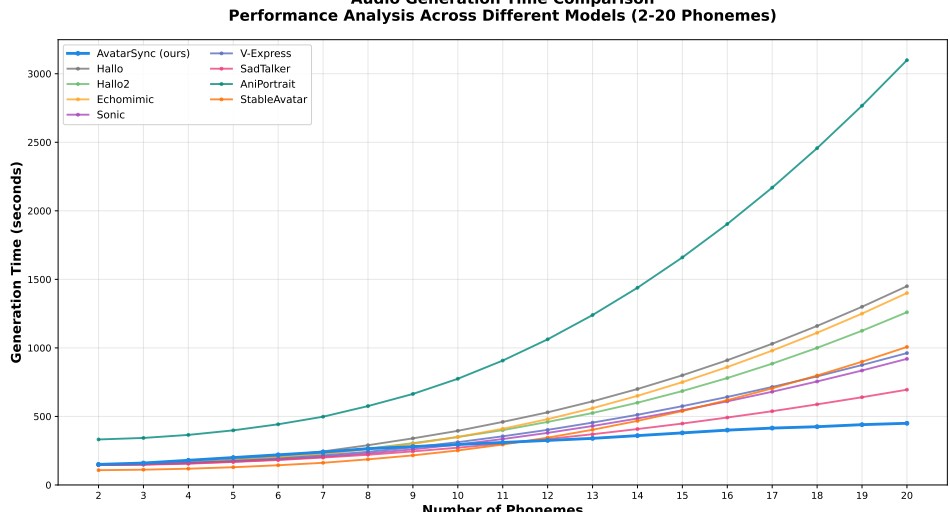

Figure 10: Generation Time Comparison. AvatarSync scales nearly linearly with phoneme count, while others exhibit exponential growth.

| Configuration | Exp. | Loss Functions | | | | Evaluation Metrics | | | |
|---|---|---|---|---|---|---|---|---|---|
| | | CE | LPIPS | Identity | Facial | FID↓ | LPIPS↓ | PSNR↑ | SSIM↑ |
| **Baseline** | 1 | ✓ | | | | 28.1361 | 0.0365 | 25.0786 | 0.8837 |
| | 2 | ✓ | ✓ | | | 16.6485 | **0.0128** | 32.5706 | 0.9615 |
| + Single Additional Loss | 5 | ✓ | | ✓ | | 16.1956 | 0.0138 | 32.0478 | 0.9620 |
| | 6 | ✓ | | | ✓ | 16.4723 | 0.0131 | 32.4699 | 0.9653 |
| | 3 | ✓ | ✓ | ✓ | | 15.6558 | 0.0151 | 31.9085 | 0.9632 |
| + Double Additional Losses | 4 | ✓ | ✓ | | ✓ | 18.1449 | 0.0162 | 31.8410 | 0.9623 |
| | 7 | ✓ | | ✓ | ✓ | **13.4429** | 0.0133 | 32.4377 | 0.9643 |
| **Full Model** | 8 | ✓ | ✓ | ✓ | ✓ | 13.8603 | 0.0136 | **33.1348** | **0.9666** |

Table 6: Ablation study on different loss function combinations. **CE**: Cross-Entropy Loss; **LPIPS**: Learned Perceptual Image Patch Similarity; **Identity**: Identity Consistency Loss; **Facial**: Facial Similarity Loss. ↓: lower is better; ↑: higher is better. Bold = best; underlined = second best per column.

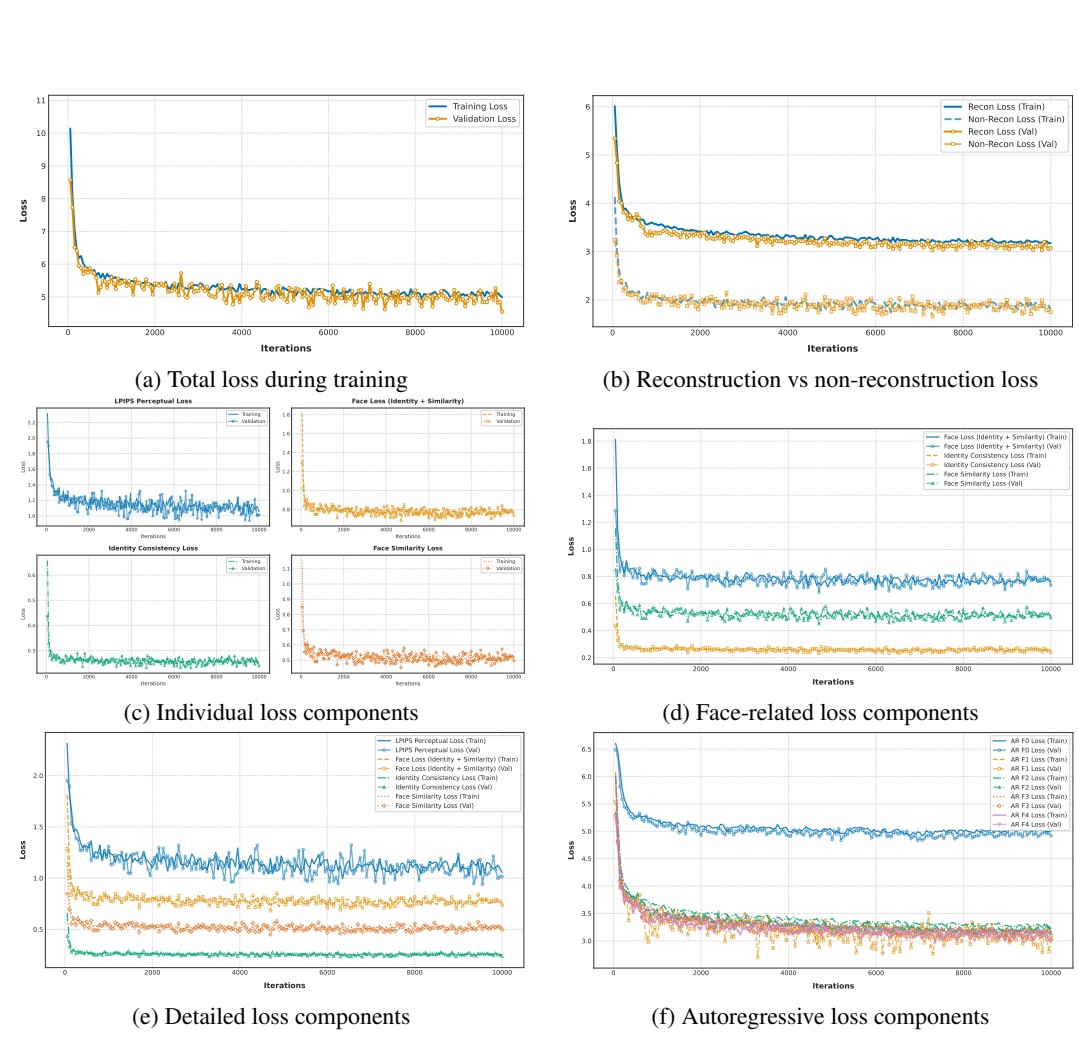

Figure 11: Training loss curves on the mixed dataset (CMLR + HDTF). The plots illustrate the convergence of various loss components over 10,000 training steps. Key metrics include reconstruction objectives, face-specific metrics, and autoregressive losses.

