# OpenReview forum: "AvatarSync: Rethinking Talking-Head Animation through Phoneme-Guided Autoregressive Perspective"
_ICLR.cc/2026/Conference — Submitted to ICLR 2026_

### Official Review · Reviewer_1bVj · 2025-10-19

**Soundness:** 3
**Presentation:** 3
**Contribution:** 2
**Rating:** 4
**Confidence:** 5

**Summary:**

This paper proposes an autoregressive generation framework for the talking-head video generation task. To alleviate the one-to-many relationship between speech and motion, as well as the flicker issue in previous autoregressive methods, the framework converts audio input into phonemes and then adopts a two-stage generation strategy: the first stage generates key frames, and the second stage generates interpolated frames between the key frames. Experiments demonstrate that the proposed method achieves promising performance.

**Strengths:**

- The paper employs an Autoregressive Method (instead of the commonly used diffusion methods). Such an attempt is relatively rare, and the method achieves excellent efficiency and quality.
- The paper conducts extensive experimental demonstrations, covering multiple aspects of metrics, which is quite comprehensive.
- The paper is written very clearly and is easy to understand.

**Weaknesses:**

- The proposed two-stage "key frame + interpolation" method lacks novelty, as this is actually an existing generation strategy (the "Two temporal res." mentioned in [1]). The authors need to clarify the core differences between this two-stage generation strategy and previous methods.
- Using phonemes may lead to information loss, such as emotion-related information and stress-related information. This may result in the lack of vividness in head movements, eye blinks, and facial expressions. The authors need to demonstrate the necessity of using phonemes (e.g., through ablation studies comparing with the commonly used audio embedding features in previous works) or provide theoretical proof that this modification is essential.
- Additionally, the paper lacks subjective metrics, such as user studies. It is recommended that the authors supplement such experiments or explain the reasons for being unable to do so.
Reference


[1] Harvey, William, et al. "Flexible diffusion modeling of long videos." Advances in neural information processing systems 35 (2022): 27953-27965.

**Questions:**

Please refer to the weakness.

---

> ### Author Response · Authors · 2025-11-25
> **Response to Reviewer 1bVj (1/2)**
>
> Dear Reviewer,
>
> Thank you for your work on our manuscript. We have studied the valuable comments carefully. We would like to clarify a few points regarding your question and hope to receive a higher score if possible. In addition, we are trying our best to revise the manuscript and hope that the corrections will meet with approval.
>
> The point to point responses to your comments are listed as follows:
>
> >[W1] I question the novelty of the proposed two-stage “key frame + interpolation” framework, arguing it resembles the existing strategy in [1], and requests a clearer explanation of how the generation pipeline fundamentally differs from prior methods.
>
> [A1] We are extremely grateful to you for pointing out this profound and critical problem. AvatarSync is primarily proposed as a novel phoneme-based autoregressive framework designed to effectively mitigate the inter-frame flickering that current diffusion-based talking-head models often suffer from. In addition, to further enhance temporal coherence and improve inference efficiency, we introduce a two-stage hierarchical generation strategy. While the idea of first modeling a sparse structure and then refining it densely may resemble prior work, there are fundamental differences. Most prior methods, such as [1], focus on modeling long videos, where keyframes are selected at roughly uniform intervals and are not explicitly tied to the semantic content of the video. In contrast, our two-stage design is a deliberate divide-and-conquer strategy, which is manifested in the following aspects:
>
> (1) We decouple semantics from visual details to ensure core task quality. The stage 1 (Keyframe Generation, FKG) focuses on semantic accuracy, and the stage 2 (Frame Interpolation) focuses on temporal coherence and visual smoothness.
>
> (2) Our two-stage design effectively cuts off remote error dependence, alleviating autoregressive error accumulation.
>
> (3) Our two-stage design improves inference efficiency by enabling a parallel pipeline for the keyframes generation and frames interpolation tasks.
>
> (4) The model features local editability, allowing users to correct an erroneous segment by regenerating only that part, not the entire video.
>
> Within the next two days, we will refine the description of Avatarsync’s two-stage generation strategy in Section 3.2 of the paper.
>
> [1] Harvey W, Naderiparizi S, Masrani V, et al. Flexible diffusion modeling of long videos[J]. Advances in neural information processing systems, 2022, 35: 27953-27965.

---

> ### Author Response · Authors · 2025-11-25
> **Response to Reviewer 1bVj (2/2)**
>
> >[W2] I question the necessity of using phoneme-based representations, due to concerns that they may discard emotion-related information and stress-related information. The authors need to demonstrate the necessity of using phonemes.
>
> [A2] We would like to clarify that the core novelty of our work lies in a phoneme-based two-stage autoregressive framework specifically designed to address two key limitations of diffusion-based approaches: (1) inter-frame flickering in the generated videos, and (2) low training and inference efficiency. We acknowledge that AvatarSync, like most talking-head generation methods, still faces challenges in modeling fine-grained emotions and stress patterns. However, this does not mean that our model is incapable of producing expressive results. We are already extending the second stage of our framework to incorporate emotional and other auxiliary cues, and have produced positive results.
>
> Regarding the necessity of using phonemes:
>
> (1) Phonemes are the smallest speech units defined according to the natural properties of spoken language. Phoneme sequences provide a compact, linguistically grounded representation, which is crucial for precise lip motion.
>
> (2) Compared with high-dimensional audio embeddings, the “many-to-one”  phoneme mapping significantly reduces the joint modeling space of text and audio, and filters out nuisance factors such as background noise. This enables more efficient encoding, more stable audio-visual alignment, and improved cross-lingual robustness.
>
> (3) By conditioning on phoneme sequences, we explicitly decouple semantics from visual detail. With this phoneme-based two-stage strategy, AvatarSync achieves better visual fidelity, motion consistency, and inference speed than existing methods, as demonstrated by the experimental results in Section 4.2. It is also worth noting that commonly used audio embeddings, in addition to speech content, often encode background sounds and noise, which can affect the performance of the first stage.
>
> (4) In Section 4.4, we further conduct an ablation study on phoneme-based representation learning. Specifically, we compare (i) a baseline that represents speech only with the original lexical tokens, and (ii) our proposed variant that maps the transcript to a small, unified phoneme collection. The results show that the compact phoneme sequence consistently improves multiple metrics of the model.
>
> In Section 3.2 of the revised paper, we will clarify the theoretical basis of the necessity of phonemes and its connection with the ablation experiment in Section 4.4.
>
> >[W3] The paper lacks subjective metrics, such as user studies.
>
> [A3] Thank you for your valuable suggestion regarding experiments. We agree that a user study on inter-frame flicker is an important component for assessing model performance. In response, we conducted a human evaluation to assess the quality of generated animations. Thirty participants rated nine state-of-the-art methods focusing on flicker, body movement, temporal coherence, and lip synchronization.
>
> **Table 3: Human evaluation results.**
>
> | Method | Flicker ↑    | Body Movement ↑ | Temporal Coherence ↑ | Lip Sync ↑    | Total ↑ |
> |--------|--------------|-----------------|----------------------|---------------|---------|
> | AvatarSync (ours) | 4.8 ± 0.41 | 3.8 ± 0.48 | 4.4 ± 0.50 | 3.6 ± 0.62 | 16.6 |
> | Sonic| 3.0 ± 0.59 | 3.8 ± 0.66 | 3.2 ± 0.76 | 4.0 ± 0.45 | 14.0 |
> | SadTalker| 4.6 ± 0.72 | 2.2 ± 0.81 | 3.8 ± 0.55 | 3.0 ± 0.45 | 13.6 |
> | StableAvatar| 3.2 ± 0.55 | 3.0 ± 0.59 | 3.2 ± 0.61 | 3.0 ± 0.74 | 12.4 |
> | EchoMimic| 2.6 ± 0.81 | 1.2 ± 0.41 | 2.0 ± 0.74 | 1.0 ± 0.26 | 6.8 |
> | Hallo| 3.2 ± 0.61 | 0.8 ± 0.48 | 0.8 ± 0.41 | 1.0 ± 0.45 | 5.8 |
> | AniPortrait| 2.8 ± 0.61 | 1.0 ± 0.53 | 1.2 ± 0.66 | 0.8 ± 0.48 | 5.8 |
> | Hallo2| 2.6 ± 0.67 | 0.8 ± 0.55 | 1.0 ± 0.53 | 1.0 ± 0.45 | 5.4 |
> | V-Express| 1.2 ± 0.48 | 0.8 ± 0.41 | 1.0 ± 0.45 | 0.8 ± 0.55 | 3.8 |
>
> As shown in the table above, AvatarSync outperforms other models in the comprehensive evaluation. Within the next two days, we will supplement the human evaluation results in Section 4 of the paper.

---

> ### Author Response · Authors · 2025-12-01
> **Response to Reviewer 1bVj**
>
> Dear Reviewers,
>
> Thank you for your careful and valuable reviews on our submission. Although the review scores and further discussion are now frozen due to the recent OpenReview incident, we would still like to honor our commitment to you. We are pleased to inform you that we have substantially revised the manuscript based on your comments.
>
> In the revised manuscript, the following major revisions have been made:
>
> (1) Further clarified the overall motivation and contribution of AvatarSync, a phoneme-guided, two-stage autoregressive talking-head framework, and better explained how it addresses two key limitations of diffusion-based approaches: inter-frame flickering in generated videos, and low inference efficiency. (See the abstract, Section 1, and Sections 3.2-3.4 for details.)
>
> (2) Strengthened the experimental section by expanding the set of baselines and evaluation metrics, adding targeted analyses and ablations (e.g., attention mechanisms and frame allocation strategies), and conducting a human evaluation with 30 participants over 9 methods, so as to more directly address the concerns raised in your reviews. Furthermore, we have provided code and video results at the following anonymous GitHub link (https://anonymous.4open.science/r/AvatarSync-HYPN/README.md). (See Sections 4.1-4.4 and Appendices A.7 and A.8 for details)
>
> (3) Improved the presentation and organization by clarifying the overall architecture and training setup, and adding clearer explanations of the roles of each stage and key component (e.g., phoneme-based representation learning (PRL) and the timestamp-aware frame allocation strategy). We also made the discussion of potential future work more explicit. (See Section 4.5, Section 5, and Appendices A.4-A.6 for details)
>
> We fully understand that you can no longer change your scores or participate in further discussion, but we sincerely appreciate the time and effort you have already invested. We hope that the updated manuscript addresses your main concerns and more clearly conveys the contributions and technical soundness of our work.
>
> Thank you again for your thoughtful feedback and for helping us improve this paper.

---

### Official Review · Reviewer_rEVM · 2025-10-27

**Soundness:** 3
**Presentation:** 3
**Contribution:** 3
**Rating:** 6
**Confidence:** 4

**Summary:**

This paper proposes AvatarSync, an autoregressive framework for talking-head animation that leverages phoneme representations and a two-stage generation strategy to produce high-quality, lip-synchronized facial videos from a single image and audio input.

**Strengths:**

（1）Structurally, the paper is well-organized, with clear explanations of the methodology, theoretical analysis of inter-frame flicker, and illustrative figures.
（2）In terms of quality, extensive experiments on two benchmark datasets demonstrate clear improvements in both quantitative metrics (e.g., FID, FVD) and qualitative results.
（3）The paper introduces a novel autoregressive framework that effectively leverages phoneme representations for talking-head generation, with the phoneme-frame causal attention mask and two-stage hierarchical generation strategy being both creative and well-motivated.

**Weaknesses:**

（1）The method primarily focuses on comparisons with strong baselines like Hallo, while insufficient attention is given to including newer approaches such as VASA-1. It also lacks direct comparisons with other autoregressive methods (e.g., VideoPoet, Teller) in terms of speed and resource consumption. Although comparisons with diffusion models demonstrate certain advantages, they fail to adequately establish its leading position within the autoregressive paradigm.（2）The experimental evaluation mainly concentrates on overall video quality and lip synchronization, while lacking specialized assessment and analysis of subtle facial expressions such as natural blinking and lip pursing.
（3）The approach relies heavily on multiple high-performance GPUs, which may limit its reproducibility for users with consumer-grade hardware. Demonstrating its performance on more accessible computing resources would be valuable.

**Questions:**

（1）How well does the model scale with longer phoneme sequences? Does performance degradation occur when generating extended videos?
（2）The paper states that the interpolation module can process different keyframe pairs in parallel. Does this imply that the entire second stage is fully parallelizable? If so, what is the theoretical speedup ratio? Please elaborate on the specific parallelization strategy and its practical benefits in real-world deployment.
（3）Have you considered incorporating more diverse scenarios in experiments, such as adding noise, to further test the model's robustness in real-world environments?
（4）The phoneme-frame causal attention mask forces each keyframe to focus solely on its corresponding phoneme. Could this strict one-to-one mapping limit the model's ability to learn coarticulation effects across adjacent phonemes? Would this limitation affect the fluency and naturalness of the resulting lip movements?
（5）The experiments involve a 4× super-resolution preprocessing step on the CMLR dataset. Is there any quantitative evidence demonstrating the specific contribution of this operation to the final model performance compared to using the original data?

---

> ### Author Response · Authors · 2025-11-25
> **Response to Reviewer rEVM (1/3)**
>
> Dear Reviewer,
>
> Thank you for your work on our manuscript. We have studied the valuable comments carefully. We would like to clarify a few points regarding your question and hope to receive a higher score if possible. In addition, we are trying our best to revise the manuscript and hope that the corrections will meet with approval.
>
> The point to point responses to your comments are listed as follows:
>
> >[W1] The method lacks comparisons with newer approaches such as VASA-1. Moreover, it fails to establish its leadership within the autoregressive paradigm, as it omits comparisons against other autoregressive methods (e.g., VideoPoet, Teller).
>
> [A1] We are extremely grateful to you for pointing out this profound and critical problem. We agree that it is better to add comparisons to newer approaches, such as VASA-1, which adopts a two-stage framework with intermediate facial motion representations followed by diffusion-based video synthesis. However, VASA-1 has not released training or evaluation code. To address your suggestion, we conduct additional experiments with AniPortrait, which is similar to VASA-1. In addition, we have added a comparison between AvatarSync and AniPortrait on CMLR and HDTF.
>
> **Table 1: Quantitative comparison on CMLR.**
>
> |Method|FID↓|FVD↓|LPIPS↓|PSNR↑|SSIM↑|ArcFace/CSIM↑|Sync-C↓|Sync-D↓|
> |------|----|----|------|-----|-----|-------------|-------|-------|
> |SadTalker|30.96|618.74|0.42|14.50|0.48|0.79|0.91|14.29|
> |V-Express|32.54|821.95|0.46|12.99|0.39|0.70|0.83|13.71|
> |AniPortrait|32.46|711.30|0.47|14.71|**0.49**|0.69|0.89|14.99|
> |Hallo|30.84|630.53|0.43|14.62|**0.49**|0.71|1.20|14.99|
> |Hallo2|35.10|570.12|0.44|14.59|**0.49**|0.75|**0.80**|14.67|
> |EchoMimic|33.09|1225.10|0.42|14.87|0.48|0.69|0.89|13.97|
> |Sonic|31.58|953.30|0.43|14.39|0.47|0.68|2.25|12.84|
> |StableAvatar|39.38|741.36|0.46|14.41|**0.49**|0.79|0.92|14.83|
> |**AvatarSync**|**25.19**|**503.29**|**0.40**|**15.07**|**0.49**|**0.85**|0.94|**10.90**|
>
> **Table 2: Quantitative comparison on HDTF.**
>
> |Method|FID↓|FVD↓|LPIPS↓|PSNR↑|SSIM↑|ArcFace/CSIM↑|Sync-C↓|Sync-D↓|
> |------|----|----|------|-----|-----|-------------|-------|-------|
> |SadTalker|37.20|316.78|0.26|18.85|0.74|0.82|5.72|11.83|
> |V-Express|40.13|779.19|0.27|18.29|0.72|0.88|5.29|12.79|
> |AniPortrait|40.19|614.01|0.28|18.52|0.72|0.88|5.61|11.30|
> |Hallo|44.48|520.72|0.25|18.94|**0.75**|0.88|5.05|11.62|
> |Hallo2|42.77|340.83|0.25|**19.04**|**0.75**|0.89|6.37|11.91|
> |EchoMimic|38.11|301.33|0.29|18.11|0.73|0.85|2.63|12.88|
> |Sonic|32.89|379.13|0.26|18.55|0.74|0.90|7.65|12.70|
> |StableAvatar|35.39|700.82|0.28|18.41|0.72|0.85|5.34|12.56|
> |**AvatarSync**|**29.72**|**260.65**|**0.24**|18.69|0.73|**0.95**|**2.57**|**10.39**|
>
> As shown in the table above, our model achieves state-of-the-art performance in six key metrics, outperforming AniPortrait. Within the next two days, we will update the new experimental table to Section 4.2 of the paper.
>
> Regarding comparisons with other autoregressive methods: Section 2.2 already discusses recent autoregressive video generators such as VideoPoet and Teller. However, these models have not released public training or evaluation code. Moreover, existing autoregressive models in the 2D talking-head animation are still at a relatively early stage. As a result, there is currently no comparable autoregressive baseline.

---

> ### Author Response · Authors · 2025-11-25
> **Response to Reviewer rEVM (2/3)**
>
> >[W2] The experimental evaluation focuses on overall video quality and lip synchronization, but lacks dedicated analysis of subtle facial expressions such as blinking and lip pursing.
>
> [A2] We are sincerely grateful for your valuable suggestion regarding experiments. The core novelty of our work lies in a phoneme-based two-stage autoregressive framework specifically designed to address two key limitations of diffusion-based approaches: (1) inter-frame flickering in the generated videos, and (2) low training and inference efficiency. Accordingly, the experimental evaluation mainly concentrates on video quality and lip synchronization. Of course, the assessment of facial expressions is also important. In response, we conducted a human evaluation to assess the quality of generated animations. Thirty participants rated nine state-of-the-art methods focusing on flicker, body movement, temporal coherence, and lip synchronization.
>
> **Table 3: Human evaluation results.**
>
> | Method | Flicker ↑  | Body Movement ↑ | Temporal Coherence ↑ | Lip Sync ↑  | Total ↑ |
> |--------|--------------|-----------------|----------------------|---------------|---------|
> | AvatarSync (ours) | 4.8 ± 0.41 | 3.8 ± 0.48 | 4.4 ± 0.50 | 3.6 ± 0.62 | 16.6 |
> | Sonic| 3.0 ± 0.59 | 3.8 ± 0.66 | 3.2 ± 0.76 | 4.0 ± 0.45 | 14.0 |
> | SadTalker| 4.6 ± 0.72 | 2.2 ± 0.81 | 3.8 ± 0.55 | 3.0 ± 0.45 | 13.6 |
> | StableAvatar| 3.2 ± 0.55 | 3.0 ± 0.59 | 3.2 ± 0.61 | 3.0 ± 0.74 | 12.4 |
> | EchoMimic| 2.6 ± 0.81 | 1.2 ± 0.41 | 2.0 ± 0.74 | 1.0 ± 0.26 | 6.8 |
> | Hallo| 3.2 ± 0.61 | 0.8 ± 0.48 | 0.8 ± 0.41 | 1.0 ± 0.45 | 5.8 |
> | AniPortrait| 2.8 ± 0.61 | 1.0 ± 0.53 | 1.2 ± 0.66 | 0.8 ± 0.48 | 5.8 |
> | Hallo2| 2.6 ± 0.67 | 0.8 ± 0.55 | 1.0 ± 0.53 | 1.0 ± 0.45 | 5.4 |
> | V-Express| 1.2 ± 0.48 | 0.8 ± 0.41 | 1.0 ± 0.45 | 0.8 ± 0.55 | 3.8 |
>
> As shown in the table above, AvatarSync outperforms other models in the comprehensive evaluation. Within the next two days, we will supplement the human evaluation results in Section 4 of the paper.
>
> >[W3] The approach relies heavily on multiple high-performance GPUs, which may limit its reproducibility for users with consumer-grade hardware. Demonstrating its performance on more accessible computing resources would be valuable.
>
> [A3] In Section 4.1, we state that training is conducted for 10,000 steps on the mixed dataset using a total of 8 NVIDIA V100 and 2 NVIDIA L20 GPUs. However, we have verified that AvatarSync can generate videos at a resolution of 512×512 on a single RTX 4090 GPU.
>
> >[Q1] How well does the model scale with longer phoneme sequences? Does performance degradation occur when generating extended videos?
>
> [A1] Thank you for this insightful question. From an architectural perspective, AvatarSync is capable of handling longer phoneme sequences. In the experiments, the reported metrics do not indicate a degradation trend with increasing video length. In real-world deployment, long audio sequences may lead to large temporal gaps between keyframes, where the quality of the interpolated frames may degrade. We plan to further improve the model’s ability to handle such cases in future work.
>
> >[Q2] The paper claims that the interpolation module can process keyframe pairs in parallel. Does this mean the second stage is parallelizable, what is the theoretical speedup, and how does this parallelization strategy benefit real-world deployment?
>
> [A2] In AvatarSync, the interpolation module can process each keyframe pair independently. Consequently, the second stage is parallelizable across keyframe pairs, and its complexity is reduced from $\mathcal{O}\big(\sum_i L_i\big)$ to $\mathcal{O}\big(\max_i L_i\big)$, where $L_i$ denotes the length of interval $i$. Assuming $T_s$ keyframes that are approximately uniformly spaced, this design yields an ideal speedup of about $T_s - 1$ compared with a naive generator. At present, interpolation within a single interval is still executed sequentially. Next, we plan to further optimize the stage by leveraging acceleration or distillation techniques. Subsequently, we will add more details about the model’s parallel processing in Section 4.2.

---

> ### Author Response · Authors · 2025-11-25
> **Response to Reviewer rEVM (3/3)**
>
> >[Q3] Have you tested the model under more diverse, noisy conditions to assess its robustness in real-world settings?
>
> [A3] We agree that the robustness of the model is crucial for real-world deployment. At present, our training mainly focuses on standard talking-head benchmarks. As a result, the current model has not been thoroughly evaluated in challenging scenarios (e.g., masks). This is because occlusions in the mouth region disrupt the reliable alignment between the phoneme sequence and visual content. To enhance robustness, we are currently training the model on more diverse data, such as CelebV-HQ, and will update the results after the training is completed.
>
> >[Q4] The phoneme-frame causal attention enforces a strict one-to-one mapping between each keyframe and its phoneme. Could this hinder learning coarticulation across adjacent phonemes and thus reduce the naturalness of lip movements?
>
> [A4] In Section 4.4, we conduct an ablation study over four different attention mechanisms, where One-to-One Attention achieves the best trade-off between frame-level fidelity and temporal coherence. In terms of coarticulation effects, which are particularly common in our Chinese dataset, we explicitly introduce special mappings for specific coarticulation patterns (e.g., two characters are jointly pronounced). Accordingly, this significantly reduces the impact of coarticulation on the model’s performance. We also are planning to further refine the model’s handling of such phenomena in future work.
>
> Regarding the naturalness of the lip motions, we would like to clarify that AvatarSync is a novel phoneme-based autoregressive framework that decouples semantic content from visual details. As a result, the interpolation module can effectively exploit temporal information to produce smooth transitions between keyframes.
>
> >[Q5] The experiments involve a 4× super-resolution preprocessing step on the CMLR dataset. Is there any quantitative evidence demonstrating the specific contribution of this operation?
>
> [A5] Our motivation for applying super-resolution to CMLR is that its native resolution is very low, which often leads to blurry lip contours and weak identity details. We acknowledge that we do not currently provide a comparison. However, AvatarSync achieves consistent improvements on HDTF, where no super-resolution preprocessing is used, indicating that the gains mainly arise from the proposed phoneme-driven two-stage design.

---

> > ### Comment · Reviewer_rEVM · 2025-11-27
> >
> > Athough authors claim some problems, I think this paper still need to improve, so I decide to remain my score.

---

> > > ### Author Response · Authors · 2025-12-01
> > > **Response to Reviewer rEVM**
> > >
> > > Dear Reviewers,
> > >
> > > Thank you for your careful and valuable reviews on our submission. Although the review scores and further discussion are now frozen due to the recent OpenReview incident, we would still like to honor our commitment to you. We are pleased to inform you that we have substantially revised the manuscript based on your comments.
> > >
> > > In the revised manuscript, the following major revisions have been made:
> > >
> > > (1) Further clarified the overall motivation and contribution of AvatarSync, a phoneme-guided, two-stage autoregressive talking-head framework, and better explained how it addresses two key limitations of diffusion-based approaches: inter-frame flickering in generated videos, and low inference efficiency. (See the abstract, Section 1, and Sections 3.2-3.4 for details.)
> > >
> > > (2) Strengthened the experimental section by expanding the set of baselines and evaluation metrics, adding targeted analyses and ablations (e.g., attention mechanisms and frame allocation strategies), and conducting a human evaluation with 30 participants over 9 methods, so as to more directly address the concerns raised in your reviews. Furthermore, we have provided code and video results at the following anonymous GitHub link (https://anonymous.4open.science/r/AvatarSync-HYPN/README.md). (See Sections 4.1-4.4 and Appendices A.7 and A.8 for details)
> > >
> > > (3) Improved the presentation and organization by clarifying the overall architecture and training setup, and adding clearer explanations of the roles of each stage and key component (e.g., phoneme-based representation learning (PRL) and the timestamp-aware frame allocation strategy). We also made the discussion of potential future work more explicit. (See Section 4.5, Section 5, and Appendices A.4-A.6 for details)
> > >
> > > We fully understand that you can no longer change your scores or participate in further discussion, but we sincerely appreciate the time and effort you have already invested. We hope that the updated manuscript addresses your main concerns and more clearly conveys the contributions and technical soundness of our work.
> > >
> > > Thank you again for your thoughtful feedback and for helping us improve this paper.

---

### Official Review · Reviewer_ERg9 · 2025-10-30

**Soundness:** 2
**Presentation:** 2
**Contribution:** 2
**Rating:** 2
**Confidence:** 4

**Summary:**

The paper proposes AvatarSync, an autoregressive (AR) framework on phoneme presentations with a two-stage pipeline : (1) Facial KeyFrame Generation (FKG) module that maps phoneme tokens (from text or audio) to sparse keyframes under Phoneme-Frame Causal Attention Mask, and (2) a tilmestep-aware, selective state-space modeling, which enables temporally coherent inference. The paper argues autoregressive framework mitigates flicker issues and improves efficiency. Extensive experiments cover performances on both CMLR (Chinese) and HDTF (English).

**Strengths:**

1.  Phoneme-conditioned keyframes and timestamp-aware interpolation is a clean split, and one-to-one causal attention is motivated and ablated.
2. The paper cross-lingual settings.
3. Attention variants and composite losses are well-documented.

**Weaknesses:**

1. Baseline Coverage
Table1 includes several known baselines, but newer strong systems (e.g., StableAvatar, HunyuanAvatar, Hallo 3) are missing.

2. Missing human evaluation and demos.
The paper presents no user study and no supplementary videos, leaving perceptual claims (realism, temporal stability, lip sync, identity) unsupported by human evidence or visual inspection. As results rely solely on automatic metrics, the basis for performance judgment is insufficient.

3. Metric weakness
Identity/facial similarity are used as losses, but test-time identity-preservation metrics are not surfaced alongside the metrics used in the main paper. Also, the paper emphasizes "responsive user experience in the real-world setting", but does not report the metrics of autoregressive efficiency. Please provide complementary identity-related metrics (e.g., ArcFace, CSIM), and efficiency-related metrics (e.g., FPS, latency, VRAM).
Table2 omits any lip-sync metrics, and across the paper LSE-C (Sync-C) is largely absent too. Please explain the omission.

4. Attribution gap (what component actually helps?)
It is unclear how much each stage contributes. While the paper ablate the attention mechanism within FKG, there is no ablation isolating the timestamp-aware adaptive strategy in the inter-frame module;  hence the relative contributions of FKG vs. the temporal module are not established. Please add a controlled ablation that (i) disables timestamps, (ii) replaces the adaptive strategy with simpler temporal heads and (iii) final version (timestamp-aware adaptive strategy).

5. Phoneme timing and coarticulation.
A one-to-one causal mask may ignore coarticulation or context span. There is no analysis of context window , lip sync or forced alignment quality.

6. Robustness gaps
Would the authors comment on whether any cases involving occlusions, accessories (e.g., hats, glasses), pronounced head-pose variation, or mouth-region artifacts (e.g., teeth, tongue) were observed? If so, a brief analysis or examples would be helpful.

**Questions:**

I would appreciate responses to the questions in the Weaknesses section.

---

> ### Author Response · Authors · 2025-11-25
> **Response to Reviewer ERg9 (1/3)**
>
> Dear Reviewer,
>
> Thank you for your work on our manuscript. We have studied the valuable comments carefully. I would like to clarify a few points regarding your question, and we hope to receive a higher score if possible. In addition, we are trying our best to revise the manuscript and hope that the corrections will meet with approval.
>
> The point to point responses to your comments are listed as follows:
>
> >[W1] Baseline Coverage Table1 includes several known baselines, but newer strong systems are missing.
>
> [A1] Following your suggestion, we have additionally included StableAvatar, which is the latest SOTA model, in our comparisons and have achieved positive results.
>
> **Table 1: Quantitative comparison on CMLR.**
>
> |Method|FID↓|FVD↓|LPIPS↓|PSNR↑|SSIM↑|ArcFace/CSIM↑|Sync-C↓|Sync-D↓|
> |------|----|----|------|-----|-----|-------------|-------|-------|
> |SadTalker|30.96|618.74|0.42|14.50|0.48|0.79|0.91|14.29|
> |V-Express|32.54|821.95|0.46|12.99|0.39|0.70|0.83|13.71|
> |AniPortrait|32.46|711.30|0.47|14.71|**0.49**|0.69|0.89|14.99|
> |Hallo|30.84|630.53|0.43|14.62|**0.49**|0.71|1.20|14.99|
> |Hallo2|35.10|570.12|0.44|14.59|**0.49**|0.75|**0.80**|14.67|
> |EchoMimic|33.09|1225.10|0.42|14.87|0.48|0.69|0.89|13.97|
> |Sonic|31.58|953.30|0.43|14.39|0.47|0.68|2.25|12.84|
> |StableAvatar|39.38|741.36|0.46|14.41|**0.49**|0.79|0.92|14.83|
> |**AvatarSync**|**25.19**|**503.29**|**0.40**|**15.07**|**0.49**|**0.85**|0.94|**10.90**|
>
> **Table 2: Quantitative comparison on HDTF.**
>
> |Method|FID↓|FVD↓|LPIPS↓|PSNR↑|SSIM↑|ArcFace/CSIM↑|Sync-C↓|Sync-D↓|
> |------|----|----|------|-----|-----|-------------|-------|-------|
> |SadTalker|37.20|316.78|0.26|18.85|0.74|0.82|5.72|11.83|
> |V-Express|40.13|779.19|0.27|18.29|0.72|0.88|5.29|12.79|
> |AniPortrait|40.19|614.01|0.28|18.52|0.72|0.88|5.61|11.30|
> |Hallo|44.48|520.72|0.25|18.94|**0.75**|0.88|5.05|11.62|
> |Hallo2|42.77|340.83|0.25|**19.04**|**0.75**|0.89|6.37|11.91|
> |EchoMimic|38.11|301.33|0.29|18.11|0.73|0.85|2.63|12.88|
> |Sonic|32.89|379.13|0.26|18.55|0.74|0.90|7.65|12.70|
> |StableAvatar|35.39|700.82|0.28|18.41|0.72|0.85|5.34|12.56|
> |**AvatarSync**|**29.72**|**260.65**|**0.24**|18.69|0.73|**0.95**|**2.57**|**10.39**|
>
> As shown in the table above, AvatarSync achieves state-of-the-art performance outperforming Stableavatar. Within the next two days, we will update the new experimental table to Section 4.2 of the paper.
>
> >[W2] Missing human evaluation and demos. Claims of the paper about realism, temporal stability, lip sync, and identity are supported only by automatic metrics, making the performance assessment insufficient.
>
> [A2] Thank you for your valuable suggestion regarding experiments. We agree that a user study on inter-frame flicker is an important component for assessing model performance. In response, we conducted a human evaluation to assess the quality of generated animations. Thirty participants rated nine methods focusing on flicker, body movement, temporal coherence, and lip synchronization.
>
> **Table 3: Human evaluation results.**
>
> | Method | Flicker ↑    | Body Movement ↑ | Temporal Coherence ↑ | Lip Sync ↑    | Total ↑ |
> |--------|--------------|-----------------|----------------------|---------------|---------|
> | AvatarSync (ours) | 4.8 ± 0.41 | 3.8 ± 0.48 | 4.4 ± 0.50 | 3.6 ± 0.62 | 16.6 |
> | Sonic| 3.0 ± 0.59 | 3.8 ± 0.66 | 3.2 ± 0.76 | 4.0 ± 0.45 | 14.0 |
> | SadTalker| 4.6 ± 0.72 | 2.2 ± 0.81 | 3.8 ± 0.55 | 3.0 ± 0.45 | 13.6 |
> | StableAvatar| 3.2 ± 0.55 | 3.0 ± 0.59 | 3.2 ± 0.61 | 3.0 ± 0.74 | 12.4 |
> | EchoMimic| 2.6 ± 0.81 | 1.2 ± 0.41 | 2.0 ± 0.74 | 1.0 ± 0.26 | 6.8 |
> | Hallo| 3.2 ± 0.61 | 0.8 ± 0.48 | 0.8 ± 0.41 | 1.0 ± 0.45 | 5.8 |
> | AniPortrait| 2.8 ± 0.61 | 1.0 ± 0.53 | 1.2 ± 0.66 | 0.8 ± 0.48 | 5.8 |
> | Hallo2| 2.6 ± 0.67 | 0.8 ± 0.55 | 1.0 ± 0.53 | 1.0 ± 0.45 | 5.4 |
> | V-Express| 1.2 ± 0.48 | 0.8 ± 0.41 | 1.0 ± 0.45 | 0.8 ± 0.55 | 3.8 |
>
> As shown in the table above, AvatarSync outperforms other models in the comprehensive evaluation.
>
> In addition, to further demonstrate the generative performance of our model, we have provided additional video results at the following anonymous GitHub link (https://anonymous.4open.science/r/AvatarSync-HYPN/README.md). Based on the video results, AvatarSync has resolved the key limitation of inter-frame flickering. Within the next two days, we will supplement the human evaluation results in Section 4 of the paper and the anonymous link in the abstract of the paper.

---

> ### Author Response · Authors · 2025-11-25
> **Response to Reviewer ERg9 (2/3)**
>
> >[W3] The paper does not report test-time identity-preservation metrics and the metrics of autoregressive efficiency. In addition, Table 2 omits lip-sync metrics and LSE-C (Sync-C) is largely missing throughout. Please clarify these omissions and add the relevant metrics.
>
> [A3] The core novelty of our work lies in a phoneme-based two-stage autoregressive framework specifically designed to address two key limitations of diffusion-based approaches: (1) inter-frame flickering in the generated videos, and (2) low training and inference efficiency. Accordingly, the experimental evaluation mainly concentrates on video quality and lip synchronization. Following prior work, we initially adopted widely used metrics such as the lip-sync metrics Sync-D.
>
> Following your advice, we have strengthened the evaluation by adding additional metrics and comparisons, including identity-related metrics (ArcFace/CSIM) and the lip-sync metrics Sync-C. The corresponding results are reported in Tabs. 1 and 2, where our model achieves state-of-the-art performance on six key metrics, outperforming existing baselines.
>
> Regarding efficiency, Sec. 4.2 already presents curves of inference time versus the number of input phonemes, comparing AvatarSync with the diffusion-based baselines and demonstrating its superior autoregressive efficiency. Follow your suggestion, we supplemented efficiency-related metrics, among which AvatarSync achieved the best performance. Subsequently, we will supplement the experimental results in Section 4.2 of the paper.
>
> **Table 4: Inference efficiency on 3.92s (8-phoneme) clips at 512×512 resolution.**
>
> |Method|Latency (s)↓|RTF↓|
> |------|-----------:|---:|
> |SadTalker|78.90|20.13|
> |V-Express|167.90|42.83|
> |AniPortrait|575.50|146.81|
> |Hallo|444.30|113.34|
> |Hallo2|427.50|109.06|
> |EchoMimic|512.70|130.79|
> |Sonic|190.40|48.57|
> |StableAvatar|187.00|47.70|
> |AvatarSync|58.00|14.80|
>
> >[W4] Attribution gap. It is unclear how much each stage contributes. Please add a controlled ablation that (i) disables timestamps, (ii) replaces the adaptive strategy with simpler temporal heads and (iii) timestamp-aware adaptive strategy.
>
> [A4] AvatarSync is a novel phoneme-based autoregressive framework whose two-stage design is an intentional strategy. By decoupling semantic content from visual details, the first stage focuses on semantic and phonetic accuracy, while the second stage is dedicated to temporal coherence and visual smoothness. In the second stage, the timestamp-aware adaptive strategy determines the number of in-between frames for each keyframe pair according to their temporal interval and the target frame rate, thereby maintaining a globally consistent frame rate and producing temporally coherent, rhythmically natural videos. Following your advice, we have added a controlled ablation of frame allocation strategies.
>
> **Table 5: Ablation of frame allocation strategies.**
>
> |Method|FID↓|FVD↓|LPIPS↓|PSNR↑|SSIM↑|ArcFace/CSIM↑|Sync-C↓|Sync-D↓|
> |------|----|----|------|-----|-----|-------------|-------|-------|
> |Random|24.82|553.39|**0.40**|**15.07**|0.49|**0.85**|1.09|14.11|
> |Fixed|**24.56**|522.32|0.41|15.04|0.49|0.84|**0.94**|14.08|
> |Dynamic (Ours)|25.19|**503.29**|**0.40**|**15.07**|0.49|**0.85**|**0.94**|**10.90**|
>
> As shown in the table above, the timestamp-aware adaptive strategy achieves the best performance on seven metrics. Subsequently, we will supplement the experimental results in Section 4.4 of the paper.

---

> ### Author Response · Authors · 2025-11-25
> **Response to Reviewer ERg9 (3/3)**
>
> >[W5] Phoneme timing and coarticulation. A one-to-one causal mask may ignore coarticulation or context span. There is no analysis of context window , lip sync or forced alignment quality.
>
> [A5] In Section 4.4, we conduct an ablation study over four different attention mechanisms with different context span. Following your suggestion, we have completed the evaluation and add additional metric: Sync-C, which will be updated in Table 2 of the paper.
>
> **Table 6: Ablation of attention mechanisms.**
>
> |ID|Setting|FID↓|FVD↓|LPIPS↓|PSNR↑|SSIM↑|Sync-C↓|
> |--|-------|----|----|------|-----|-----|-------|
> |1|Non-Causal|**15.63**|287.95|0.07±0.01|**24.41**|0.86|**1.21**|
> |2|Causal Acc.|19.47|210.25|0.07±0.01|23.35|0.86|1.24|
> |3|Lim. Hist.|19.56|**186.68**|0.07±0.01|23.83|0.86|1.32|
> |4|One-to-One|17.11|189.24|0.07±0.01|24.14|0.86|1.32|
>
> As shown in the table above, One-to-One Attention achieves the best trade-off between frame-level fidelity, lip synchronization and temporal coherence.
>
> In terms of coarticulation effects, which are particularly common in our Chinese dataset, we explicitly introduce special mappings for specific coarticulation patterns (e.g., two characters are jointly pronounced). Accordingly, this significantly reduces the impact of coarticulation on the model’s performance. We also are planning to further refine the model’s handling of such phenomena in future work.
>
> >[W6] Robustness gaps Would the authors comment on whether any cases involving occlusions, accessories, pronounced head-pose variation, or mouth-region artifacts were observed? If so, a brief analysis or examples would be helpful.
>
> [A6] We agree that the robustness of the model is crucial for real-world deployment.  At present, our training mainly focuses on standard talking-head benchmarks. As a result, the current model does not yet perform reliably in challenging scenarios (e.g., masks). This is because occlusions in the mouth region disrupt the reliable alignment between the phoneme sequence and visual content. To address this limitation, we are currently training the model on more diverse data, such as CelebV-HQ.

---

> ### Author Response · Authors · 2025-12-01
> **Response to Reviewer ERg9**
>
> Dear Reviewers,
>
> Thank you for your careful and valuable reviews on our submission. Although the review scores and further discussion are now frozen due to the recent OpenReview incident, we would still like to honor our commitment to you. We are pleased to inform you that we have substantially revised the manuscript based on your comments.
>
> In the revised manuscript, the following major revisions have been made:
>
> (1) Further clarified the overall motivation and contribution of AvatarSync, a phoneme-guided, two-stage autoregressive talking-head framework, and better explained how it addresses two key limitations of diffusion-based approaches: inter-frame flickering in generated videos, and low inference efficiency. (See the abstract, Section 1, and Sections 3.2-3.4 for details.)
>
> (2) Strengthened the experimental section by expanding the set of baselines and evaluation metrics, adding targeted analyses and ablations (e.g., attention mechanisms and frame allocation strategies), and conducting a human evaluation with 30 participants over 9 methods, so as to more directly address the concerns raised in your reviews. Furthermore, we have provided code and video results at the following anonymous GitHub link (https://anonymous.4open.science/r/AvatarSync-HYPN/README.md). (See Sections 4.1-4.4 and Appendices A.7 and A.8 for details)
>
> (3) Improved the presentation and organization by clarifying the overall architecture and training setup, and adding clearer explanations of the roles of each stage and key component (e.g., phoneme-based representation learning (PRL) and the timestamp-aware frame allocation strategy). We also made the discussion of potential future work more explicit. (See Section 4.5, Section 5, and Appendices A.4-A.6 for details)
>
> We fully understand that you can no longer change your scores or participate in further discussion, but we sincerely appreciate the time and effort you have already invested. We hope that the updated manuscript addresses your main concerns and more clearly conveys the contributions and technical soundness of our work.
>
> Thank you again for your thoughtful feedback and for helping us improve this paper.

---

### Official Review · Reviewer_7vpw · 2025-11-07

**Soundness:** 2
**Presentation:** 3
**Contribution:** 3
**Rating:** 4
**Confidence:** 3

**Summary:**

The paper focuses on the inter-frame flickering and slow inference problems of existing GAN and diffusion models, and proposes an autoregressive framework that leverages a many-to-one mapping from text/audio to phonemes and a two-stage strategy to enhance temporal coherence and visual smoothness. Experimental results show improved temporal stability and efficiency of the proposed method.

**Strengths:**

1. The phoneme tokenization is inspiring, and the representation is easy to follow.
2. Empirical results show improved video quality, temporal coherence, and efficiency, which support the claims of the paper.

**Weaknesses:**

1. The paper only considers one reference image for the framework, while in practice, multiple reference images with different facial angles might be provided. How will the framework scale to this case? And will the diversity among reference images introduce noise into the tokens?
2. Humans are sensitive to inter-frame flicker, but there is a lack of user study in the experiments.
3. Providing synthesized video examples is recommended to further support the claims of the paper.

**Questions:**

Please answer the questions in the weakness.

---

> ### Author Response · Authors · 2025-11-25
> **Response to Reviewer 7vpw**
>
> Dear Reviewer,
>
> Thank you for your work on our manuscript. We have studied the valuable comments carefully. I would like to clarify a few points regarding your question, and we hope to receive a higher score if possible. In addition, we are trying our best to revise the manuscript and hope that the corrections will meet with approval.
>
> The point to point responses to your comments are listed as follows:
>
> >[W1] I am concerned about the model’s ability to scale to a multi-reference setting and about whether the diversity of reference views might introduce noisy into the token representations.
>
> [A1] We are extremely thankful to you for pointing out this important problem. However, we would like to clarify that the core novelty of our work lies in a phoneme-based two-stage autoregressive framework specifically designed to address two key limitations of diffusion-based approaches: (1) inter-frame flickering in the generated videos, and (2) low training and inference efficiency. The scenario of using multiple reference images with diverse facial angles is beyond the scope of the present work, and we plan to explicitly extend and evaluate AvatarSync in future work. It is worth noting that AvatarSync can naturally be extended to multiple reference views, such as an entire talking-head video.
>
> (1)When a reference video and target speech audio are provided, we first sample frames from the reference video and select the frame that best represents the speaker’s identity as a global reference frame, which helps ensure consistent appearance in the generated video. Additionally, we match the phoneme sequence to the frames in the reference video and extract the facial landmarks from matching frames. In the first stage, the model conditions on the phoneme sequence, the phoneme-level facial landmarks, and the global reference frame to generate a sparse set of speech-synchronized keyframes. In the second stage, conditioned on the full audio, the interpolation module performs temporal interpolation and detail refinement between these keyframes, synthesizing intermediate frames. This process ultimately produces a temporally continuous, identity-consistent talking-head video.
>
> (2)We agree that the diversity among reference images introduces noise into the tokens. However, the robustness of the model is crucial for real-world deployment. At present, our training mainly focuses on standard talking-head benchmarks such as CMLR and HDTF. To further enhance robustness, we are currently training the model on more diverse data, such as CelebV-HQ, and will update the results after the training is completed.
>
> >[W2] Humans are sensitive to inter-frame flicker, but there is a lack of user study in the experiments.
>
> [A2] Thank you for your valuable suggestion regarding experiments. We agree that a user study on inter-frame flicker is an important component for assessing model performance. In response, we conducted a human evaluation to assess the quality of generated animations. Thirty participants rated nine methods focusing on flicker, body movement, temporal coherence, and lip synchronization.
>
> **Table 1: Human evaluation results.**
>
> | Method | Flicker ↑    | Body Movement ↑ | Temporal Coherence ↑ | Lip Sync ↑    | Total ↑ |
> |--------|--------------|-----------------|----------------------|---------------|---------|
> | AvatarSync (ours) | 4.8 ± 0.41 | 3.8 ± 0.48 | 4.4 ± 0.50 | 3.6 ± 0.62 | 16.6 |
> | Sonic| 3.0 ± 0.59 | 3.8 ± 0.66 | 3.2 ± 0.76 | 4.0 ± 0.45 | 14.0 |
> | SadTalker| 4.6 ± 0.72 | 2.2 ± 0.81 | 3.8 ± 0.55 | 3.0 ± 0.45 | 13.6 |
> | StableAvatar| 3.2 ± 0.55 | 3.0 ± 0.59 | 3.2 ± 0.61 | 3.0 ± 0.74 | 12.4 |
> | EchoMimic| 2.6 ± 0.81 | 1.2 ± 0.41 | 2.0 ± 0.74 | 1.0 ± 0.26 | 6.8 |
> | Hallo| 3.2 ± 0.61 | 0.8 ± 0.48 | 0.8 ± 0.41 | 1.0 ± 0.45 | 5.8 |
> | AniPortrait| 2.8 ± 0.61 | 1.0 ± 0.53 | 1.2 ± 0.66 | 0.8 ± 0.48 | 5.8 |
> | Hallo2| 2.6 ± 0.67 | 0.8 ± 0.55 | 1.0 ± 0.53 | 1.0 ± 0.45 | 5.4 |
> | V-Express| 1.2 ± 0.48 | 0.8 ± 0.41 | 1.0 ± 0.45 | 0.8 ± 0.55 | 3.8 |
>
> As shown in the table above, AvatarSync outperforms other models in the comprehensive evaluation. Within the next two days, we will supplement the human evaluation results in Section 4 of the paper.
>
> >[W3] Providing synthesized video examples is recommended to further support the claims of the paper.
>
> [A3] We greatly appreciate your helpful suggestion. Following your advice, we have provided additional video results at the following anonymous GitHub link (https://anonymous.4open.science/r/AvatarSync-HYPN/README.md). Based on the video results, AvatarSync has successfully resolved the key limitation of inter-frame flickering in generated videos. Subsequently, we will supplement the anonymous link in the abstract of the paper.

---

> ### Author Response · Authors · 2025-12-01
> **Response to Reviewer 7vpw**
>
> Dear Reviewers,
>
> Thank you for your careful and valuable reviews on our submission. Although the review scores and further discussion are now frozen due to the recent OpenReview incident, we would still like to honor our commitment to you. We are pleased to inform you that we have substantially revised the manuscript based on your comments.
>
> In the revised manuscript, the following major revisions have been made:
>
> (1) Further clarified the overall motivation and contribution of AvatarSync, a phoneme-guided, two-stage autoregressive talking-head framework, and better explained how it addresses two key limitations of diffusion-based approaches: inter-frame flickering in generated videos, and low inference efficiency. (See the abstract, Section 1, and Sections 3.2-3.4 for details.)
>
> (2) Strengthened the experimental section by expanding the set of baselines and evaluation metrics, adding targeted analyses and ablations (e.g., attention mechanisms and frame allocation strategies), and conducting a human evaluation with 30 participants over 9 methods, so as to more directly address the concerns raised in your reviews. Furthermore, we have provided code and video results at the following anonymous GitHub link (https://anonymous.4open.science/r/AvatarSync-HYPN/README.md). (See Sections 4.1-4.4 and Appendices A.7 and A.8 for details)
>
> (3) Improved the presentation and organization by clarifying the overall architecture and training setup, and adding clearer explanations of the roles of each stage and key component (e.g., phoneme-based representation learning (PRL) and the timestamp-aware frame allocation strategy). We also made the discussion of potential future work more explicit. (See Section 4.5, Section 5, and Appendices A.4-A.6 for details)
>
> We fully understand that you can no longer change your scores or participate in further discussion, but we sincerely appreciate the time and effort you have already invested. We hope that the updated manuscript addresses your main concerns and more clearly conveys the contributions and technical soundness of our work.
>
> Thank you again for your thoughtful feedback and for helping us improve this paper.

---

### Official Review · Reviewer_AupQ · 2025-11-11

**Soundness:** 3
**Presentation:** 3
**Contribution:** 2
**Rating:** 4
**Confidence:** 4

**Summary:**

This paper presents AvatarSync, a novel framework for audio-driven talking-head animation that proposes a paradigm shift away from the prevailing GAN and diffusion-based approaches. The authors identify critical limitations in existing methods: GANs often suffer from visual artifacts and instability, while diffusion models, despite their high fidelity, are plagued by slow inference speeds and a characteristic inter-frame flicker that breaks temporal consistency.
To address these issues, AvatarSync reframes the task as a phoneme-guided autoregressive sequence generation problem, inspired by recent successes of large language models in video synthesis. The core of the method is a clever two-stage generation strategy designed to balance quality and efficiency:
Facial Keyframe Generation (FKG): An autoregressive model generates a sparse set of high-quality facial keyframes conditioned on phoneme sequences extracted from the input audio/text. A key innovation here is the "Phoneme-Frame Causal Attention Mask," which enforces a strict alignment between phonemes and their corresponding visual frames, ensuring precise lip synchronization.
Inter-Frame Interpolation: A second, lightweight module then fills in the intermediate frames between the generated keyframes. This module uses a timestamp-aware strategy to ensure the final video is temporally coherent, smooth, and computationally inexpensive to render.
The authors provide extensive experimental validation on the CMLR and HDTF datasets, demonstrating that AvatarSync achieves state-of-the-art results across multiple metrics, including visual fidelity (FID, FVD), lip-sync accuracy (Sync-D), and identity preservation. Crucially, the method is shown to be significantly faster than diffusion-based competitors and qualitatively free from the inter-frame flicker problem.

**Strengths:**

1. The paper does not merely propose an incremental improvement but "rethinks" the entire problem. By identifying the fundamental limitations of GANs and diffusion models for this specific task and proposing a coherent autoregressive alternative, the authors make a significant conceptual contribution. The framing is clear, well-motivated, and timely.
2. The FKG + Interpolation architecture is a highly intelligent solution to the quality-efficiency trade-off. It leverages the power of a large autoregressive model for the most critical parts of the animation (the keyframes) while delegating the more repetitive task of in-betweening to a faster, specialized module. This design is a key reason for the method's success.
3. A major strength of AvatarSync is its ability to generate temporally stable videos. The autoregressive generation of keyframes inherently builds upon previous frames, and the interpolation further ensures smoothness. The flicker visualization in Figure 5(b) is a powerful and convincing piece of evidence that directly showcases the superiority of this approach over diffusion models in eliminating distracting artifacts.

**Weaknesses:**

1. The paper's core motivation rests heavily on the premise that diffusion-based methods are inherently slow due to their iterative denoising process. While this is true for traditional DDPM/DDIM samplers, this argument is becoming outdated. Recent advancements in diffusion model distillation have enabled real-time, one-step generation. For instance, [1] demonstrate a diffusion-based approach that achieves real-time performance. The paper's efficiency claims would be much stronger if they were contextualized against these modern, distilled diffusion models, rather than just standard, slow samplers. Without this discussion, the claimed efficiency advantage feels overstated.
2. The paper introduces a two-stage generation process but fails to compare against other highly relevant and state-of-the-art two-stage methods. A critical omission is AniPortrait [2], a very popular and powerful framework that also employs a two-stage approach (first generating keyframes/poses and then rendering the video). Given the conceptual similarity in using a two-stage pipeline to tackle the problem, a direct comparison—or at least a detailed discussion—is essential to properly situate AvatarSync's contribution. Without it, it is difficult to assess whether the novelty and performance gains come from the autoregressive formulation itself or simply from adopting a two-stage strategy.

Reference:
[1].Guo, Hanzhong, et al. "Real-time One-Step Diffusion-based Expressive Portrait Videos Generation." arXiv preprint arXiv:2412.13479 (2024).
[2].Wei, Huawei, Zejun Yang, and Zhisheng Wang. "Aniportrait: Audio-driven synthesis of photorealistic portrait animation." arXiv preprint arXiv:2403.17694 (2024).

**Questions:**

See the weakness.

---

> ### Author Response · Authors · 2025-11-25
> **Response to Reviewer AupQ (1/2)**
>
> Dear Reviewer,
>
> Thank you for your work on our manuscript. We have studied the valuable comments carefully. I would like to clarify a few points regarding your question, and we hope to receive a higher score if possible. In addition, we are trying our best to revise the manuscript and hope that the corrections will meet with approval.
>
> The point to point responses to your comments are listed as follows:
>
> >[W1] I question the paper’s efficiency-centered motivation, as it is based on diffusion being inherently slow and ignores modern real-time distilled diffusion models, so the claimed advantage may be overstated.
>
> [A1] We are sincerely grateful for your valuable suggestion regarding experiments. AvatarSync is primarily proposed as a novel phoneme-based autoregressive framework designed to effectively mitigate the inter-frame flickering that diffusion-based talking-head models often suffer from. In addition, to further enhance temporal coherence and improve inference efficiency, we introduce a two-stage hierarchical generation strategy. As shown by the generation speed comparison in Sec. 4.2, AvatarSync exhibits a significant advantage over other models in inference efficiency and provides a promising paradigm for further optimization. Combined with the development of acceleration techniques for autoregressive models summarized in Sec. 2.2, these results suggest that autoregressive talking-head generation holds strong potential for real-time applications. It is important to emphasize that this does not negate the recent progress made by diffusion models in improving inference speed.
>
> Following your valuable suggestion, we have compared AvatarSync with modern, distilled diffusion models. After surveying [1], [2], and [3], we found that only [3] provides publicly available code. We therefore conducted a comparison between AvatarSync and Ditto[3] under the same conditions.
>
> **Table 1: Comparison between AvatarSync and Ditto.**
>
> |Method|Audio Duration (s)|Latency (s)↓|RTF↓|
> |------|------------------|-----------:|---:|
> |AvatarSync|2.11|55|26.07|
> |Ditto|2.11|23|10.90|
> |AvatarSync|22.89|68|2.97|
> |Ditto|22.89|77|3.36|
>
> As shown in the table, Ditto attains lower latency than AvatarSync on short clips.  However, for longer audio, AvatarSync outperforms Ditto in inference speed, because its interpolation module processes keyframe pairs in parallel. More importantly, as discussed in Sec. 2.2, AvatarSync remains compatible with distillation and acceleration techniques for further real-time improvement. Within the next two days, we will add the comparison with Ditto to Figure 4 in the paper.
>
> [1] Guo H, Yi H, Zhou D, et al. Real-time One-Step Diffusion-based Expressive Portrait Videos Generation[J]. arXiv preprint arXiv:2412.13479, 2024.
>
> [2] Yu H, Wang Z, Pan Y, et al. LLIA--Enabling Low-Latency Interactive Avatars: Real-Time Audio-Driven Portrait Video Generation with Diffusion Models[J]. arXiv preprint arXiv:2506.05806, 2025.
>
> [3] Li T, Zheng R, Yang M, et al. Ditto: Motion-space diffusion for controllable realtime talking head synthesis[C]//Proceedings of the 33rd ACM International Conference on Multimedia. 2025: 9704-9713.

---

> ### Author Response · Authors · 2025-11-25
> **Response to Reviewer AupQ (2/2)**
>
> >[W2] I question whether the performance gains truly come from the proposed autoregressive design, as the paper omits comparison with state-of-the-art two-stage methods like AniPortrait, making it unclear how much is due to the formulation rather than the two-stage strategy.
>
> [A2] We are extremely grateful to you for pointing out this profound and critical problem. AniPortrait is indeed a representative two-stage model. Following your suggestion, we have included AniPortrait in our comparisons. In addition, as shown in the table 1 and 2, AvatarSync achieves state-of-the-art performance in six key metrics on both the CMLR and HDTF datasets, outperforming AniPortrait. Within the next two days, we will update the new experimental table to Section 4.2 of the paper.
>
> **Table 2: Quantitative comparison on CMLR.**
>
> |Method|FID↓|FVD↓|LPIPS↓|PSNR↑|SSIM↑|ArcFace/CSIM↑|Sync-C↓|Sync-D↓|
> |------|----|----|------|-----|-----|-------------|-------|-------|
> |SadTalker|30.96|618.74|0.42|14.50|0.48|0.79|0.91|14.29|
> |V-Express|32.54|821.95|0.46|12.99|0.39|0.70|0.83|13.71|
> |AniPortrait|32.46|711.30|0.47|14.71|**0.49**|0.69|0.89|14.99|
> |Hallo|30.84|630.53|0.43|14.62|**0.49**|0.71|1.20|14.99|
> |Hallo2|35.10|570.12|0.44|14.59|**0.49**|0.75|**0.80**|14.67|
> |EchoMimic|33.09|1225.10|0.42|14.87|0.48|0.69|0.89|13.97|
> |Sonic|31.58|953.30|0.43|14.39|0.47|0.68|2.25|12.84|
> |StableAvatar|39.38|741.36|0.46|14.41|**0.49**|0.79|0.92|14.83|
> |**AvatarSync**|**25.19**|**503.29**|**0.40**|**15.07**|**0.49**|**0.85**|0.94|**10.90**|
>
> **Table 3: Quantitative comparison on HDTF.**
>
> |Method|FID↓|FVD↓|LPIPS↓|PSNR↑|SSIM↑|ArcFace/CSIM↑|Sync-C↓|Sync-D↓|
> |------|----|----|------|-----|-----|-------------|-------|-------|
> |SadTalker|37.20|316.78|0.26|18.85|0.74|0.82|5.72|11.83|
> |V-Express|40.13|779.19|0.27|18.29|0.72|0.88|5.29|12.79|
> |AniPortrait|40.19|614.01|0.28|18.52|0.72|0.88|5.61|11.30|
> |Hallo|44.48|520.72|0.25|18.94|**0.75**|0.88|5.05|11.62|
> |Hallo2|42.77|340.83|0.25|**19.04**|**0.75**|0.89|6.37|11.91|
> |EchoMimic|38.11|301.33|0.29|18.11|0.73|0.85|2.63|12.88|
> |Sonic|32.89|379.13|0.26|18.55|0.74|0.90|7.65|12.70|
> |StableAvatar|35.39|700.82|0.28|18.41|0.72|0.85|5.34|12.56|
> |**AvatarSync**|**29.72**|**260.65**|**0.24**|18.69|0.73|**0.95**|**2.57**|**10.39**|
>
> It is worth noting that the generation paradigm in AvatarSync differs substantially from AniPortrait and similar models. In AniPortrait, the first stage predicts a per-frame sequence of 3D facial meshes and head poses for the entire video, and the second stage uses a diffusion model conditioned on this sequence to synthesize the video. In contrast, AvatarSync first leverages phoneme information to generate a sparse set of keyframes, and then performs interpolation based on these keyframes and temporal structure to produce the video. Specifically, the key characteristics of AvatarSync’s two-stage design are reflected in the following aspects:
>
> (1)We decouple semantics from visual details to ensure core task quality. The stage 1 focuses on semantic accuracy, and the stage 2 focuses on temporal coherence and visual smoothness.
>
> (2)Our two-stage design effectively severs cuts off remote error dependence,  alleviating autoregressive error accumulation.
>
> (3)Our two-stage design improves inference efficiency by enabling a parallel pipeline for the keyframes generation and frames interpolation tasks.
>
> (4)The model features local editability, allowing users to correct an erroneous segment by regenerating only that part, not the entire video.

---

> ### Author Response · Authors · 2025-12-01
> **Response to Reviewer AupQ**
>
> Dear Reviewers,
>
> Thank you for your careful and valuable reviews on our submission. Although the review scores and further discussion are now frozen due to the recent OpenReview incident, we would still like to honor our commitment to you. We are pleased to inform you that we have substantially revised the manuscript based on your comments.
>
> In the revised manuscript, the following major revisions have been made:
>
> (1) Further clarified the overall motivation and contribution of AvatarSync, a phoneme-guided, two-stage autoregressive talking-head framework, and better explained how it addresses two key limitations of diffusion-based approaches: inter-frame flickering in generated videos, and low inference efficiency. (See the abstract, Section 1, and Sections 3.2-3.4 for details.)
>
> (2) Strengthened the experimental section by expanding the set of baselines and evaluation metrics, adding targeted analyses and ablations (e.g., attention mechanisms and frame allocation strategies), and conducting a human evaluation with 30 participants over 9 methods, so as to more directly address the concerns raised in your reviews. Furthermore, we have provided code and video results at the following anonymous GitHub link (https://anonymous.4open.science/r/AvatarSync-HYPN/README.md). (See Sections 4.1-4.4 and Appendices A.7 and A.8 for details)
>
> (3) Improved the presentation and organization by clarifying the overall architecture and training setup, and adding clearer explanations of the roles of each stage and key component (e.g., phoneme-based representation learning (PRL) and the timestamp-aware frame allocation strategy). We also made the discussion of potential future work more explicit. (See Section 4.5, Section 5, and Appendices A.4-A.6 for details)
>
> We fully understand that you can no longer change your scores or participate in further discussion, but we sincerely appreciate the time and effort you have already invested. We hope that the updated manuscript addresses your main concerns and more clearly conveys the contributions and technical soundness of our work.
>
> Thank you again for your thoughtful feedback and for helping us improve this paper.

---

### Author Response · Authors · 2025-12-02
**Summary for AC Consideration**

Dear Area Chair,

Thank you for taking the time to evaluate our submission under the exceptional circumstances this year. This comment provides **(i) a concise overview of the paper, and (ii) a summary of the revisions made in direct response to reviewer feedback.**

**1.Paper overview: core motivation and contribution**

Current diffusion-based talking-head approaches suffer from two major limitations: **(1) inter-frame flicker in generated videos, and (2) slow inference efficiency,** which are hindering their practical deployment.

To this end, we introduce AvatarSync, a novel phoneme-guided autoregressive framework that generates talking-head animations from a single reference image, driven by text or audio. In addition, to further improve temporal coherence and inference efficiency, we introduce a two-stage hierarchical generation strategy that decouples phoneme semantics from visual dynamics. The first stage, Facial Keyframe Generation (FKG), constructs phoneme-to-visual alignment and produces a sparse set of keyframes, by leveraging the many-to-one mapping from text or audio to phonemes. In the second stage, we propose a timestamp-aware interpolation module to adaptively allocate and synthesize intermediate frames between keyframes. This design mitigates error accumulation, supports localized editing, and enables parallel inference for improved efficiency.

Finally, we evaluate AvatarSync on both a Chinese benchmark (CMLR) and an English benchmark (HDTF), and it consistently outperforms recent GAN-based and diffusion-based methods across visual, identity, and lip-sync metrics. Additionally, we conducted a human evaluation in which participants rated AvatarSync highest on flicker, body movement, temporal coherence, and competitive on lip synchronization, indicating robust and consistent performance. Furthermore, we have provided code and video results at the following anonymous GitHub link (https://anonymous.4open.science/r/AvatarSync-HYPN/README.md).

**2.Main revisions in response to the five reviews**

Following the reviewers’ suggestions, we implemented the following substantial revisions to the manuscript:

**(1) Stronger baselines and metrics.** We added StableAvatar and AniPortrait as baselines on both datasets and expanded the metrics to include ArcFace/CSIM and Sync-C, providing a more complete view. In addition, AvatarSync is state-of-the-art or highly competitive across all metrics.

**(2) Human evaluation.** We conducted a user study with 30 participants evaluating 9 methods on flicker, body movement, temporal coherence, and lip synchronization. AvatarSync achieves the highest overall score, with strong scores across all dimensions and low variance in scores, which indicates robust and consistent performance.

**(3) Quantitative efficiency analysis.** We reported quantitative runtime results for all baselines in terms of latency and real-time factor (RTF), and additionally included a direct comparison with Ditto, which is a modern and distilled diffusion model. The results show that our model’s latency exhibits a near-linear relationship with the input phoneme count, outperforming other models.

**(4) Clarifying novelty and component roles.** We further clarified the overall motivation and contributions of AvatarSync, by providing clearer descriptions of the architecture. Additionally, we added targeted analyses and ablations on phoneme-based representation learning (PRL), attention masking, and the timestamp-aware frame allocation strategy, showing how each stage and key module contributes to the final performance.

**(5) Code and demo release.** To further demonstrate the generative performance of our model, we have provided code and additional video results at the following anonymous GitHub link (https://anonymous.4open.science/r/AvatarSync-HYPN/README.md).

We hope this brief summary is helpful for your assessment of both the core ideas of AvatarSync and the revisions we have made in response to the reviews. Supported by solid empirical evidence and analysis, the current version presents a coherent and well-substantiated modeling paradigm for talking-head animation. As video generation is currently a very active research area in AI, we believe that AvatarSync makes a meaningful contribution that may stimulate further discussion in the community. We would be honored if our work could be presented and discussed at the conference, and we respectfully appreciate your consideration.

Thank you again for your time and consideration.

Sincerely,

Submission 18589 Authors

---

### Meta-Review · Area_Chair_KDc4 · 2026-01-01

**Summary:**

The reviewers broadly agree that the paper is clearly written and proposes an interesting phoneme-guided autoregressive framing with a two-stage “keyframe generation + interpolation” pipeline. However, the suggested Reject decision is primarily informed by recurring concerns that (i) the novelty and positioning relative to prior two-stage generation strategies is not sufficiently clarified, (ii) the efficiency motivation/claims are weaker without convincing contextualization against modern accelerated/distilled diffusion approaches and without consistent wins across regimes, and (iii) the evaluation in the initial submission lacked key evidence for perceptual quality and real-world robustness (human study/demos, identity/lip-sync/efficiency reporting, robustness and scope beyond single-reference settings). While the rebuttal reports additions (stronger baselines/metrics, a human study, efficiency tables, and code/videos), reviewers’ core “what truly drives the gains” and “how strong/novel is the contribution” concerns remain only partially resolved based on the discussion record provided.

**Reviewer Concerns:**

### Concerns addressed (partially or fully) by rebuttal

- Missing baselines / stronger comparisons: Multiple reviewers requested inclusion of newer/stronger systems and relevant two-stage methods (e.g., AniPortrait, StableAvatar). The rebuttal states these were added with expanded quantitative tables.

- Missing metrics (identity + lip-sync) and incomplete reporting: Requests for test-time identity metrics (e.g., ArcFace/CSIM) and additional lip-sync metrics (e.g., Sync-C) were addressed in the rebuttal via new tables including these metrics.

- Lack of subjective evidence (human study) and demos: Reviewers noted that perceptual claims (flicker/temporal coherence/realism) were unsupported without human evaluation and visual demos. The rebuttal reports a 30-participant human evaluation across 9 methods and provides an anonymous code/video link.

- Efficiency reporting: Reviewers asked for concrete efficiency metrics (latency/RTF/FPS/VRAM). The rebuttal provides latency/RTF tables and discussion of how runtime scales with phoneme length, and includes a comparison to a distilled diffusion baseline (Ditto).

### Concerns still outstanding (decision-critical)

- Novelty and attribution of gains remain unclear: Despite additional comparisons and some ablations, reviewers’ central concern persists: it is still not fully established (from the discussion record alone) whether improvements stem from the autoregressive phoneme formulation versus the two-stage design choice (keyframe + interpolation) that resembles prior strategies.

- Efficiency claim remains regime-dependent: The rebuttal comparison indicates a distilled diffusion baseline can be faster on short clips, with AvatarSync potentially advantaged on longer audio under certain parallelization assumptions. This weakens a blanket “diffusion is inherently slow” motivation and suggests the efficiency narrative remains insufficiently settled.

- Scope and robustness limitations remain largely unaddressed experimentally: Reviewers raised concerns about (i) extending beyond single-reference inputs (multi-view/multi-reference) and (ii) robustness to occlusions/accessories/pose/mouth artifacts. The rebuttal acknowledges these as future work or limitations, rather than providing convincing empirical coverage.

- Phoneme representation risks and coarticulation analysis: Reviewers questioned potential information loss (emotion/stress) and whether strict one-to-one masking limits coarticulation modeling. The rebuttal provides partial discussion and attention ablations, but the discussion record does not show a dedicated forced-alignment quality analysis, and reviewer questions about expressiveness beyond lip motion remain.

**Reviewer Scores:**

Reviewer AupQ: 4 (as submitted)

Reviewer 7vpw: 4 (as submitted)

Reviewer ERg9: 2 (as submitted)

Reviewer rEVM: 6 (as submitted), and the reviewer explicitly states they remain at their score in a follow-up comment.

Reviewer 1bVj: 4 (as submitted)

---

### Decision · Program_Chairs · 2026-01-26

Reject